# Distinct neural encoding of glimpsed and masked speech in multitalker situations

**Vinay S Raghavan**[1,2], **James O'Sullivan**[1,2], **Stephan Bickel**[3,4,5], **Ashesh D. Mehta**[3,4], **Nima Mesgarani**[1,2]*

**1** Department of Electrical Engineering, Columbia University, New York, New York, United States of America, **2** Zuckerman Mind Brain Behavior Institute, Columbia University, New York, New York, United States of America, **3** The Feinstein Institutes for Medical Research, Northwell Health, Manhasset, New York, United States of America, **4** Department of Neurosurgery, Zucker School of Medicine at Hofstra/Northwell, Hempstead, New York, United States of America, **5** Department of Neurology, Zucker School of Medicine at Hofstra/Northwell, Hempstead, New York, United States of America

* nima@ee.columbia.edu

**Data Availability Statement:** There are restrictions to the availability of this dataset due to the protection of human subjects who participated in this study. The data that support the findings of this study are available upon request from the

## Abstract

Humans can easily tune in to one talker in a multitalker environment while still picking up bits of background speech; however, it remains unclear how we perceive speech that is masked and to what degree non-target speech is processed. Some models suggest that perception can be achieved through glimpses, which are spectrotemporal regions where a talker has more energy than the background. Other models, however, require the recovery of the masked regions. To clarify this issue, we directly recorded from primary and non-primary auditory cortex (AC) in neurosurgical patients as they attended to one talker in multitalker speech and trained temporal response function models to predict high-gamma neural activity from glimpsed and masked stimulus features. We found that glimpsed speech is encoded at the level of phonetic features for target and non-target talkers, with enhanced encoding of target speech in non-primary AC. In contrast, encoding of masked phonetic features was found only for the target, with a greater response latency and distinct anatomical organization compared to glimpsed phonetic features. These findings suggest separate mechanisms for encoding glimpsed and masked speech and provide neural evidence for the glimpsing model of speech perception.

## Introduction

Humans can converse in complex, multitalker acoustic environments in which one talker is of interest and one or more background talkers can be ignored [1]. One of the main challenges in these environments is stream segregation, or the isolation of a sound stream from the complex mixture [2]. It is thought that this process relies on local spectrotemporal structure, including onset and offset synchrony, harmonicity, and fundamental frequency continuity [3]. Due to the natural temporal variations of speech, multitalker speech inherently contains simpler moments when the target talker is louder than the background (glimpsed target) and more difficult moments when non-target speech is louder than the target (masked target or glimpsed

Scientific Platforms team at the Zuckerman Institute of Columbia University (rc@zi.columbia.edu). The code for pre-processing iEEG signals (https://github.com/Naplib/Naplib) and extracting stimulus features (https://github.com/naplab/naplib-python) is available online. The individual quantitative observations underlying the data summarized in Figs 2–6 and S2 are available at https://zenodo.org/record/7859760 (DOI: 10.5281/zenodo.7859760).

**Funding:** This work was supported by the National Institutes of Health (NIH), National Institute on Deafness and Other Communication Disorders (NIDCD) (DC014279 to NM). The funders had no role in the study design, data collection and analysis, decision to publish, or preparation of the manuscript.

**Competing interests:** The authors have declared that no competing interests exist

**Abbreviations:** AAD, auditory attention decoding; AC, auditory cortex; HG, Heschl's gyrus; iEEG, intracranial electroencephalography; MEG, magnetoencephalography; mSTG, middle STG; PA, posterior–anterior; pSTG, posterior STG; sEEG, stereotactic EEG; SNR, signal-to-noise ratio; STG, superior temporal gyrus; SVD, singular value decomposition; TRF, temporal response function.

non-target) [4]. These variations make stream segregation particularly challenging because they may require different means of handling the glimpsed and masked portions of speech for target and non-target talkers. Nevertheless, humans can reliably comprehend the target and ignore the background over a range of signal-to-noise ratios (SNRs). At the same time, many studies have demonstrated behavioral evidence for non-target speech processing, including informational masking [5–7], priming [8–11], and own-name recognition [1,12,13]. It remains unclear how attention influences the neural encoding of target and non-target speech and how this encoding depends on whether speech is glimpsed or masked.

Previous neuroimaging studies suggest a well-defined boundary between target and non-target speech processing. Studies of attentional speech processing in auditory cortex (AC) demonstrated the preferential encoding of task-relevant speech in non-primary regions, including superior temporal gyrus (STG) [14,15], while task-invariant responses were demonstrated in primary regions, including Heschl's gyrus (HG) [16–18]. In particular, O'Sullivan and colleagues [17] demonstrated that STG sites respond to the target talker, irrespective of how much the target is masked. Further, non-invasive studies investigating the encoding of linguistic features in multitalker speech demonstrated that the encoding of lexical and semantic information is restricted to attended speech [19,20]. In agreement with the filter model of attention [21], these studies suggest that the acoustic mixture is processed early, and attention works like a filter to block non-target acoustics and form a selective acoustic representation of target speech from which linguistic information can be extracted. The weakness of this perspective lies in its inability to explain the processing of non-target speech as seen in informational masking [5–7], priming [8–11], and own-name recognition [1,12,13].

Another unresolved issue is how a complete representation of target speech, unaffected by the degree of acoustic overlap, can be formed despite the fluctuating level of masking at the listener's periphery. A recent magnetoencephalography (MEG) study showed that masked acoustic edges are restored for both talkers in a multitalker environment with a small early effect of attention, suggesting that acoustic stream segregation occurs rapidly in AC [22]. However, how this acoustic segregation facilitates the perception of energetically masked phonetic information remains unclear. One possibility posits that we take advantage of glimpses, spectrotemporal regions in which a talker has more energy than the background. It is even hypothesized that glimpses alone are sufficient to support speech understanding [23–25]. Another view supports the notion that speech is perceived in glimpses, but additional mechanisms are needed to recover masked regions of speech to aid perception [24–28]. Furthermore, behavioral evidence for non-target speech processing has been attributed to perceptual glimpsing of the non-target talker [25]. Distinguishing between these two views requires additional evidence beyond behavioral data, for example, by directly examining the neural encoding of glimpsed and masked features of target and non-target speech in AC.

To gain deeper insight into the differential representation of target and non-target speech in a multitalker environment, we directly measured neural activity during a two-talker, co-located speech perception task using intracranial electroencephalography (iEEG) in HG and STG of neurosurgical patients. To test the role of glimpsing on continuous, naturalistic speech perception, we investigated the representational properties of glimpsed and masked acoustic and phonetic features using predictive linear models, shedding light on how attention influences the encoding of speech features in AC. This revealed a hierarchy of features increasing in latency and attentional modulation, including the attentionally invariant encoding of glimpsed speech and the temporally and anatomically distinct encoding of masked speech, providing neuroimaging evidence for the glimpse-plus-mask version of the glimpsing model in continuous speech.

## Results

We recorded bilateral iEEG responses from 7 subjects undergoing clinical treatment for epilepsy while they listened to a co-located, two-talker speech mixture. Two subjects had high-density electrocorticographic grids implanted over their left temporal lobe, including coverage of STG, and one of these subjects also had a stereotactic EEG (sEEG) depth electrode implanted in left HG. The remaining 5 subjects had sEEG depth electrodes implanted bilaterally, with varying amounts of coverage over the left and right AC. Fig 1B shows the AC electrodes from all subjects displayed on an average brain and their corresponding effect size (Cohen's D) [29] of the high-gamma response to speech versus silence. We restricted our analysis to these speech-responsive electrodes (Cohen's D > 0.2), including 66 and 59 electrodes located in HG and STG, respectively.

Subjects listened to stories read by a male and female speaker, first in isolation (data not analyzed here), then mixed at 0 dB with no spatial separation. The multitalker task was split into 4 blocks, and the subjects were instructed to attend to one talker (target) and ignore the other talker (non-target) at the beginning of each block. The stories were intermittently paused, and the subjects were instructed to repeat the last sentence of the target talker to ensure engagement with the task. The behavioral performance for all subjects was high (mean = 90%, SD = 8%, minimum = 80%).

We quantified the neural encoding properties of target and non-target speech by predicting neural responses from time-lagged features of the stimulus using mass-univariate multivariable linear regression models known as temporal response functions (TRFs) [30,31]. Here, we use "prediction" to refer to the modeling of unseen neural responses using stimulus features.

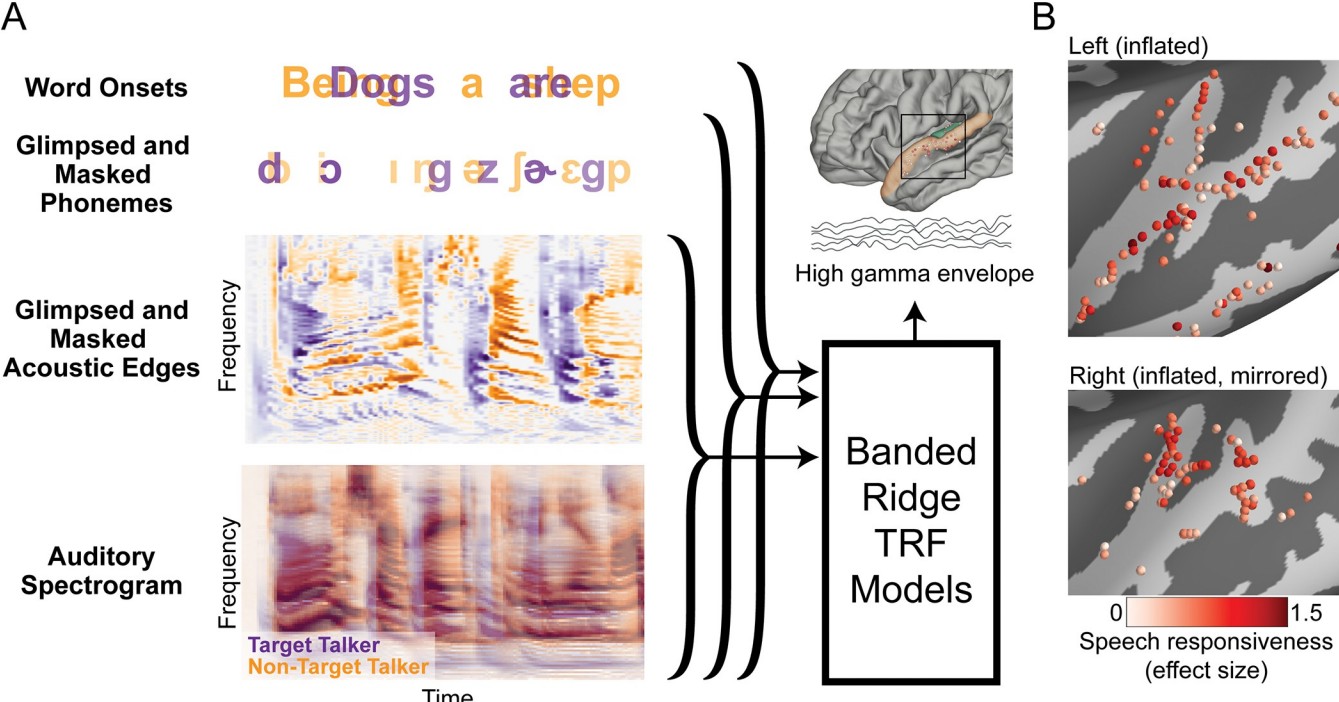

**Fig 1. Analysis framework and neural recording sites.** (A) Hierarchy of stimulus features and grouping into baseline models used to predict high-gamma neural responses using TRFs trained with banded ridge regression. The acoustic baseline model includes all spectrograms and acoustic edges. The phonetic feature baseline model includes all acoustic features, phoneme onsets, and phonetic features. The word onset baseline model includes all acoustic features, phoneme onsets, phonetic features, and word onsets. (B) Electrode coverage in auditory cortex over the inflated left (top) and right (bottom) hemispheres, showing effect size (Cohen's D) of responses to speech vs. silence. TRF, temporal response function.

These models are particularly useful because they characterize neural responses to continuous stimuli through an interpretable, linear, time-invariant system. As seen in Fig 1A, the neural responses, defined as the high-gamma band (70 to 150 Hz) envelope, were extracted and predicted from each set of stimulus features using banded ridge regression models in which each feature had a uniquely determined regularization parameter [32]. First, baseline models were trained using all relevant features. Then, null models were trained with the feature-of-interest removed. The performance of each model was determined by computing the Fisher-transformed Pearson correlation, $z$, between the model prediction and the true neural responses [33]. Thus, the degree of encoding of each feature was assessed using a one-sided, paired $t$ test between the baseline and null model correlations for the sets of electrodes in HG and STG. The significance of this improvement in correlation was tested with a one-sided hierarchical bootstrap to account for the non-independence of electrodes from the same subject [34]. The reported $p$-values were obtained from this bootstrap and provide a stricter criterion for significance than the $p$-values obtained directly from the $t$ test. Additionally, the averaged time courses of the power in the TRFs within each region are computed. Significant time points of the TRF are determined as being significantly greater ($p < 0.05$) than the corresponding time points of shuffled-feature TRFs using the hierarchical bootstrap [34]. The timing of the peak of this averaged TRF is taken as the latency of feature encoding. Together, these analyses provide insight into which features are encoded for each talker, where in AC this encoding appears, and when this encoding occurs in primary and non-primary regions of AC.

## Target acoustic edges are enhanced and masked acoustic edges are recovered with a delay

To investigate how different regions of human AC jointly respond to the different acoustic features of each talker and the mixture, we trained an acoustic baseline model comprising the spectrograms, glimpsed acoustic edges, and masked acoustic edges of each talker and the mixture, as seen in Fig 2B. Unlike spectrograms, which combine additively, rising acoustic edges, defined as the half-wave rectified temporal derivative of the spectrogram, can experience negative interference. For example, one talker may get louder as the other gets softer. In this scenario, there would be no change in the mixture envelope and, therefore, no mixture edge, despite the presence of a rising edge in the talker getting louder. Therefore, glimpsed edges are defined as a talker's edges also present in the mixture, i.e., element-wise *min(talker, mixture)*, and all mixture edges are considered glimpsed. Masked edges are then defined as the edges of a talker that exceed the edges of the mixture, i.e., element-wise *max(talker—mixture, 0)*, such that the glimpsed and masked edges sum to the original talker edges (Fig 2A). While the mixture spectrogram and glimpsed edges can be obtained from the equivalent representations of each talker, including all representations allows us to determine whether this information is best represented as the mixture and if encoding of a talker occurs above and beyond the mixture. We jointly characterized these features because neural responses in STG have been shown to primarily respond to amplitude change rather than absolute amplitude [35], and these representations have been shown to explain non-redundant portions of the neural response [19,36]. This characterization will also establish a clear comparison with MEG evidence for the distinct encoding of these acoustic features [22].

First, we predicted the high-gamma envelopes from the acoustic features and measured the reduction in prediction correlation when the target, non-target, and mixture spectrograms (Fig 2B, bottom) were removed from the predictors. As seen at the bottom of Fig 2C, the difference in correlation between the baseline and null models for electrodes in HG shows that the mixture spectrogram is strongly represented ($t_{(65)} = 6.20$, $p < 0.001$), and the peak of the

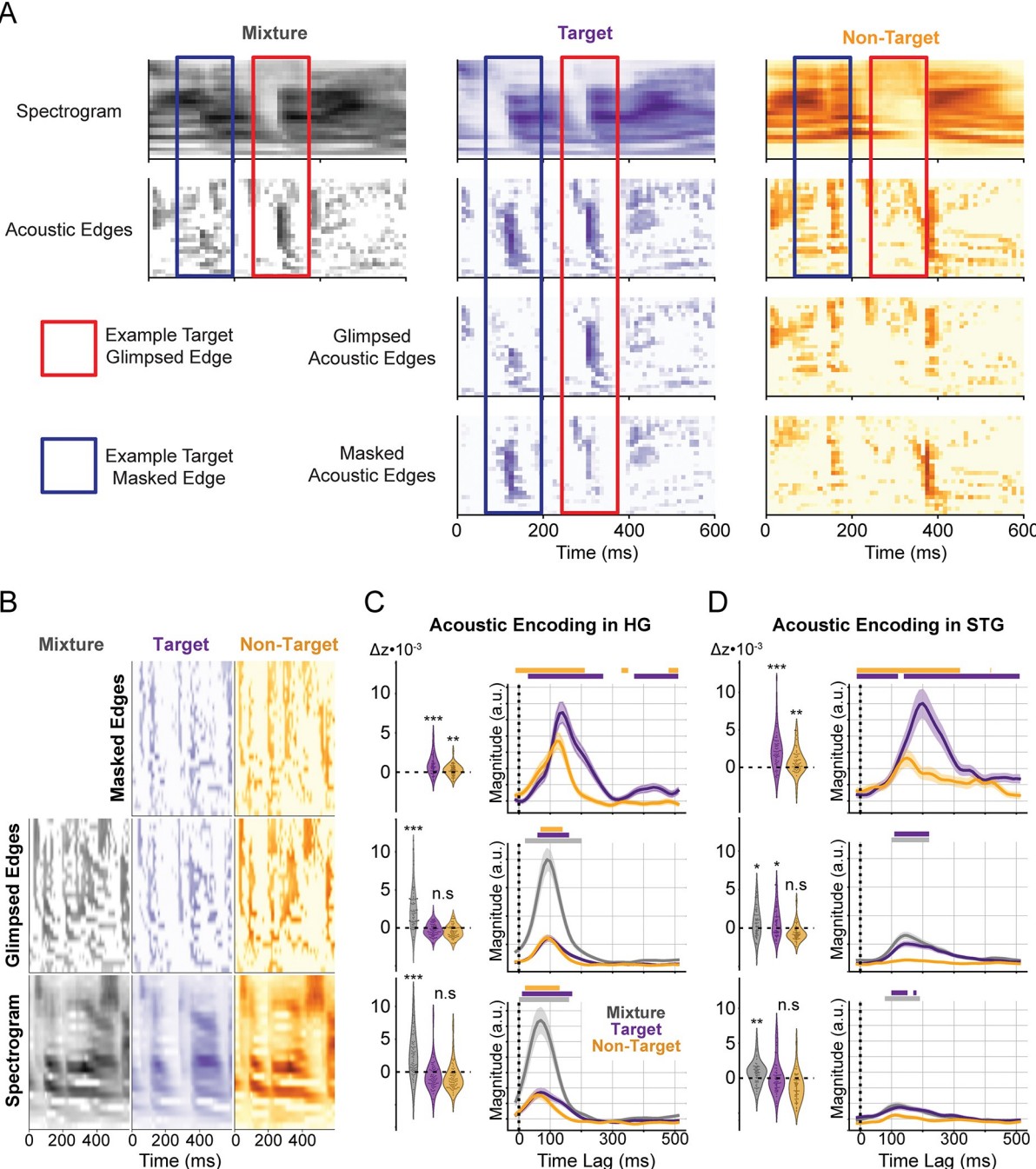

**Fig 2. Spectrogram, glimpsed edge, and masked edge features and encoding in auditory cortex.** (A) Example spectrograms (top) and extracted acoustic edges (upper middle), glimpsed acoustic edges (lower middle), and masked acoustic edges (bottom) provided for the acoustic mixture (left), target talker (center), and non-target talker (right). Blue boxes highlight an example masked acoustic edge in the target talker. This edge is absent from the acoustic edges of the mixture while being present in the acoustic edges of the target talker in isolation. Red boxes highlight the example glimpsed acoustic edge in the target talker. This edge is present in the acoustic edges of the mixture and the acoustic edges of the target talker in isolation. The presence and absence of energy in the non-target talker help illustrate why these edges are masked and glimpsed, respectively. (B) Example spectrograms (bottom), glimpsed acoustic edges (middle), and masked acoustic edges (top) of the mixture (left), target talker (center), and non-target talker (right) used in the encoding model. The prediction correlation improvement (change in Fisher-corrected correlation, one dot per electrode) and average TRF (mean ± SEM) are shown for electrodes in HG (C) and STG (D). TRF magnitudes use the same y-axis across regions, but not across features. Significant model improvements (***: $p < 0.001$, **: $p < 0.01$, *: $p < 0.05$) and TRF time steps (horizontal bars: $p < 0.05$) determined via hierarchical bootstrap. The underlying data can be found at https://zenodo.org/record/7859760. HG, Heschl's gyrus; STG, superior temporal gyrus; TRF, temporal response function.

average TRF of HG electrodes shows the encoding occurs at a latency of 70 ms. As expected, neither the target ($t_{(65)} = -0.84$, $p = 0.71$) nor the non-target ($t_{(65)} = -3.56$, $p = 0.97$) talker spectrograms were separately encoded in HG. While the individual talker TRFs contain significant sections, they align closely with the mixture TRF, suggesting that they capture the same information. Similarly, as seen at the bottom of Fig 2D, the mixture spectrogram is also significantly represented in STG ($t_{(58)} = 4.00$, $p = 0.0046$) at a latency of 120 ms. Neither the target ($t_{(58)} = 1.84$, $p = 0.089$) nor the non-target ($t_{(58)} = -3.77$, $p = 0.9993$) talker spectrograms were significantly encoded throughout STG. However, target spectrogram encoding is approaching significance, and some sites show a change in prediction correlation comparable to the mixture spectrogram, indicating some enhancement of the target spectrogram may occur at specific sites but not throughout the region.

Next, we measured the reduction in prediction correlation when the target, non-target, and mixture glimpsed edges (Fig 2B, middle) were removed from the predictors. We find that the mixture edges are strongly represented in HG ($t_{(65)} = 6.63$, $p < 0.001$) at a latency of 95 ms, while encoding of the target ($t_{(65)} = 0.083$, $p = 0.48$) and non-target ($t_{(65)} = -2.42$, $p = 0.96$), glimpsed edges were not observed in HG (Fig 2C, middle). As seen in the middle of Fig 2D, we find that the glimpsed target edges are encoded in STG ($t_{(58)} = 3.18$, $p = 0.031$) in addition to the mixture edges ($t_{(58)} = 3.44$, $p = 0.011$), both with latencies of 145 ms. We did not observe encoding of the glimpsed non-target edges in STG ($t_{(58)} = -5.96$, $p = 1.0$).

Last, we measured the reduction in prediction correlation when the target and non-target masked edges (Fig 2B, top) were removed from the predictors to determine whether acoustic features that were absent from the mixture could be recovered. As seen at the top of Fig 2C, the masked acoustic edges of both talkers are represented in HG with an effect of attention (target: $t_{(65)} = 5.13$, $p < 0.001$; non-target: $t_{(65)} = 3.45$, $p = 0.0014$). The TRFs of these representations peak at 135 and 125 ms—a delay of 40 and 30 ms relative to the latency of mixture edge encoding—for the target and non-target, respectively. Similarly, as seen at the top of Fig 2D, the masked acoustic edges of both talkers are represented in STG with a larger effect of attention (target: $t_{(58)} = 5.43$, $p < 0.001$; non-target: $t_{(58)} = 3.91$, $p = 0.0042$). The latency of masked target edges is 195 ms—a full 50 ms later than the latency of the glimpsed target edges.

Overall, we find that spectrogram information is encoded from the mixture, while the preferential encoding of the target talker in STG primarily occurs through the enhancement of acoustic edges. This enhancement occurs for both glimpsed and masked edges. Further, we find that masked edges for both talkers are encoded in HG with a delay relative to the glimpsed edges and with increasing attentional modulation from HG to STG. These results agree with MEG evidence suggesting that stream segregation is predominantly reflected in edge processing [19] and masked edges are restored with a delay [22].

## Neural responses are best predicted from glimpses with target-to-masker threshold of −4 dB

Next, we examined the encoding of speech features beyond acoustic signals by splitting the phonetic features of each phoneme, including place, manner, voice, and vowel features (S1 Fig), into glimpsed and masked representations and measuring their encoding properties. The glimpsed and masked phonetic features were quantified based on their glimpse ratio, a metric measuring the proportion of time-frequency components of the phoneme's spectrogram greater than the background spectrogram by a given threshold (Fig 3B). This metric was chosen because it represents the average value of the ideal binary mask of each phoneme [37] and was shown to be a good predictor of ASR performance and speech intelligibility [25]. To generate the glimpsed representation, the glimpse ratio of each phoneme was multiplied by the

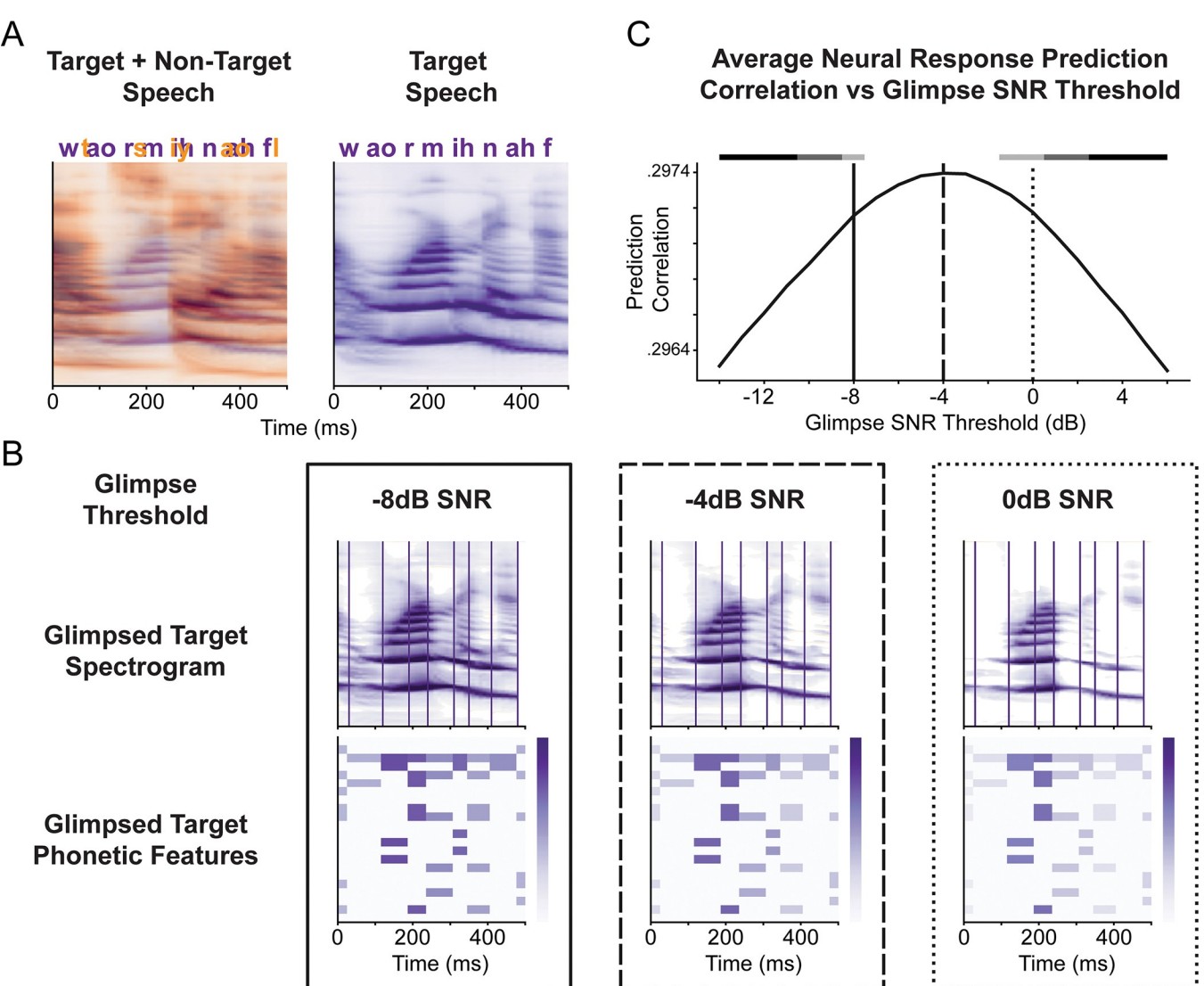

**Fig 3. Example of glimpsing for different SNR thresholds. (A)** Example target and non-target spectrograms and phonemes are shown together (left), along with the target spectrogram and phonemes alone (right). **(B)** Glimpsed phonetic feature computation for different glimpse SNR thresholds. Outlines of each SNR correspond to the vertical lines in panel **(C)**. The glimpsed target spectrogram is shown for different glimpse SNR thresholds with vertical lines indicating phoneme onsets (top). Glimpse-encoded phonetic features are shown to reveal how the SNR threshold influences glimpse values (bottom). **(C)** Average neural response prediction correlation for electrodes in HG and STG as a function of glimpse SNR thresholds. Vertical lines indicate SNR values in panel **(B)**, with a dashed line indicating the optimal SNR threshold of −4 dB. Horizontal bars above indicate significant differences between the −4 dB condition (white: ns, light gray: $p < 0.05$, dark gray: $p < 0.01$, black: $p < 0.001$). The underlying data can be found at https://zenodo.org/record/7859760. HG, Heschl's gyrus; SNR, signal-to-noise ratio; STG, superior temporal gyrus.

binary phonetic features of that phoneme. Similarly, to generate the masked representation, the mask ratio of each phoneme, i.e., 1 −glimpse ratio, was multiplied by the binary phonetic features of that phoneme. By this definition, the glimpsed and masked phonetic representations sum to the original binary phonetic features.

To investigate how HG and STG represent the glimpsed and masked phonetic features of each talker, we trained a phonetic feature baseline model comprising the acoustic features (Fig 2B), the glimpsed and masked phoneme onsets, and the glimpsed and masked phonetic features (Fig 4A). These phonetic features were added to the acoustic features to model how much phonetic information is encoded beyond acoustic information, as it has been shown

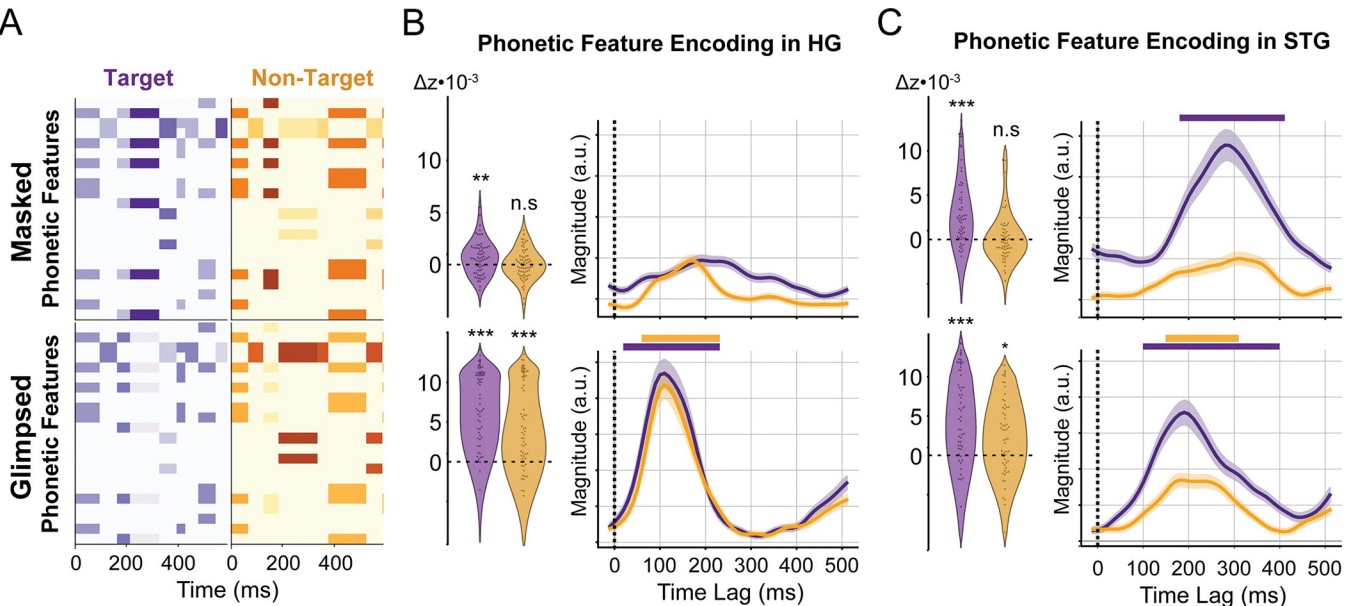

**Fig 4. Glimpsed and masked phonetic features and encoding in auditory cortex.** (**A**) Example glimpsed (bottom) and masked (top) phonetic features of the target (left) and non-target (right) talkers. The prediction correlation improvement and average TRF of electrodes in HG (**B**) and STG (**C**). TRF magnitudes are comparable across region, but not across features. Significance model improvements (***: $p < 0.001$, **: $p < 0.01$, *: $p < 0.05$) and TRF time steps (horizontal bars: $p < 0.05$) determined via hierarchical bootstrap. The underlying data can be found at https://zenodo.org/record/7859760. HG, Heschl's gyrus; STG, superior temporal gyrus; TRF, temporal response function.

that acoustic edge encoding may account for phonetic feature encoding beyond the spectrogram [36]. Additionally, phoneme onsets were included to account for the timing information contained in phonetic features. Therefore, we characterized the encoding of glimpsed and masked phonetic features separately from acoustic and timing information.

First, we sought to determine how much speech could be distorted while still being optimally classified with clean speech. We define the glimpse SNR threshold as the threshold by which each time-frequency bin of a talker must exceed the background to be considered glimpsed. Models of speech perception from glimpses alone indicate an SNR threshold of 0 dB, i.e., speech must be louder than the background to be considered glimpsed. In contrast, models of speech perception leveraging glimpsed and masked regions indicate an SNR threshold of −5 dB, i.e., speech can be up to 5 dB quieter than the background to be considered glimpsed [25]. Additionally, psychophysical data suggest that 0 dB may be a valid threshold, although a threshold closer to −6 dB is more likely [37]. As seen in Fig 3A and 3B, this threshold changes which parts of the spectrogram are considered to be glimpsed and, thus, changes the glimpse ratio of each phoneme.

We determined which glimpse SNR threshold for dividing the glimpsed and masked phonetic feature representations was representative of neural encoding by finding the threshold which optimized neural response prediction correlations. This was done by training phonetic feature baseline models using different SNR thresholds and measuring the average neural response prediction correlation as a function of the SNR threshold. We found that a glimpse SNR threshold of −4 dB was optimal for predicting neural responses across HG and STG (Fig 3C). While a threshold of −4 dB produced the largest prediction correlations, thresholds of −2 dB to −7 dB did not result in prediction correlations significantly less than those produced by the −4 dB threshold (bootstrap $p > 0.05$). All other thresholds produced significant decreases in the prediction correlation, with thresholds above 2 dB and below −10 dB

producing highly significant decreases (bootstrap $p < 0.001$). The optimal SNR threshold of −4 dB was used to divide glimpsed and masked phonetic features for all further analyses.

## Glimpsed phonetic features are encoded for both talkers with increasing attentional modulation

We then measured the reduction in prediction correlation when the target and non-target glimpsed phonetic features (Fig 4A, bottom) were removed from the predictors. As seen in the bottom row of Fig 4B, this revealed that glimpsed phonetic encoding in HG occurs robustly for both the target ($t_{(65)} = 11.44$, $p < 0.001$) and non-target ($t_{(65)} = 7.64$, $p < 0.001$) talkers and with a similar latency of 110 ms. As seen in the bottom row of Fig 4C, we found that glimpsed phonetic encoding in STG is also significant and peaks at 190 ms and 180 ms for the target and non-target talkers, respectively. However, glimpsed phonetic encoding shows a stronger effect of attention in STG than in HG, with enhanced encoding for the target over the non-target, as indicated by the model improvement (target: $t_{(58)} = 6.83$, $p < 0.001$; non-target: $t_{(58)} = 4.09$, $p = 0.015$) and the magnitude of TRF weights. These results support the notion of glimpsing as a low-level perceptual process that initially occurs invariant to attention.

## Masked phonetic features are selectively encoded for the target talker

Next, we measured the reduction in prediction correlation when the target and non-target masked phonetic features (Fig 4A, top) were removed from the predictors. As seen in the top row of Fig 4B, masked phonetic features were significantly encoded in HG for the target ($t_{(65)} = 3.26$, $p < 0.01$) but not the non-target ($t_{(65)} = 0.09$, $p = 0.48$) talker. Masked phonetic encoding in HG peaked at 190 ms; however, none of the time steps of this TRF were determined to be significant. Since TRF time steps will only be significant if TRF magnitudes are larger across the distribution of HG electrodes, this indicates that the encoding latency differs between sites in HG. As seen in the top row of Fig 4C, we again found significant encoding of masked phonetic features in STG for only the target ($t_{(58)} = 5.72$, $p < 0.001$) and not the non-target ($t_{(58)} = 0.30$, $p = 0.46$) talker. The masked target phonetic TRF in STG indicates an encoding latency of 285 ms. The attentional selectivity of masked phonetic encoding is in contrast to glimpsed phonetic and masked edge encoding observed for both talkers in both HG and STG.

## Glimpsed and masked phonetic features are encoded with distinct temporal and anatomical properties

We then investigated the temporal and anatomical encoding properties of glimpsed and masked phonetic features by comparing their encoding latency and anatomical organization in STG. First, we found differences in the latency of target phonetic feature encoding. From the average TRFs, we observed the latency of glimpsed and masked target phonetic feature encoding in HG to be 110 ms and 190 ms, respectively. Similarly, we found the latency of glimpsed and masked target phonetic feature encoding in STG to be 190 ms and 285 ms, respectively. To further quantify this difference, we computed the latency of glimpsed and masked target phonetic feature encoding for the significant electrodes in AC (Fig 5A). We found that electrodes encoding masked phonetic features had a significantly longer latency than electrodes encoding glimpsed phonetic features (Wilcoxon rank-sum test: $p < 0.001$; median glimpsed latency: 150 ms; median masked latency: 245 ms). We then computed the difference in latency for all electrodes that significantly encoded both glimpsed and masked phonetic features, finding that masked phonetic features are encoded with a significantly greater latency than glimpsed phonetic features in this population (mean difference = 53 ms;

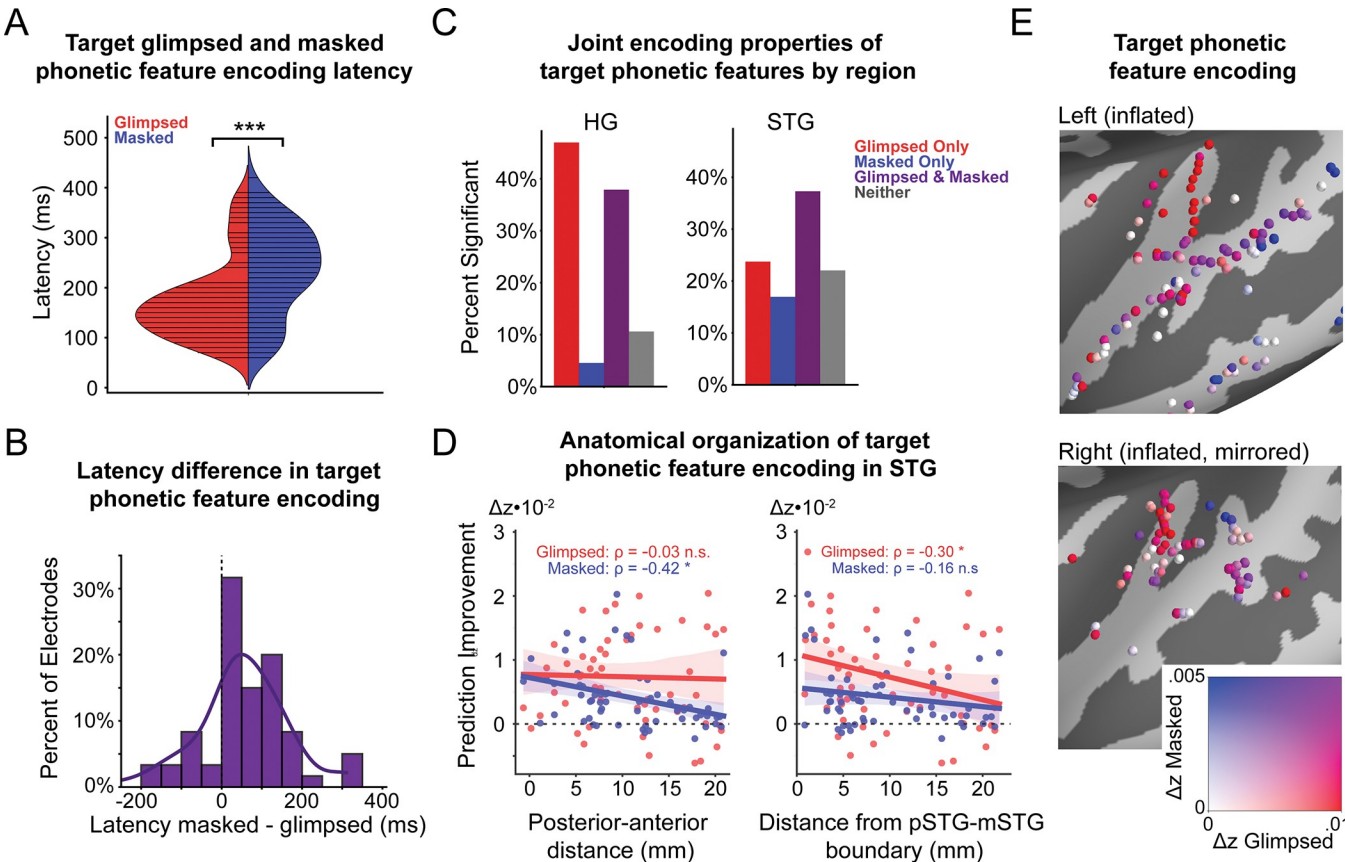

**Fig 5. Distinct temporal and anatomical encoding properties of glimpsed and masked phonetic features.** (**A**) Violin plot of the latency of all electrodes in AC that show a significant encoding of glimpsed (red) and masked (blue) target phonetic features. A significant difference is observed between the latencies of glimpsed and masked phonetic feature encoding across AC (Wilcoxon rank-sum test: $p < 0.001$; median glimpsed latency: 150 ms; median masked latency: 245 ms). (**B**) The difference in target phonetic encoding latency for electrodes in AC that encode both glimpsed and masked target phonetic features (mean difference = 64 ms). Masked phonetic features are encoded significantly later than glimpsed phonetic features over all electrodes, which encode both glimpsed and masked phonetic features (one-sample $t$ test: $t_{(61)} = 3.97$, $p < 0.001$; hierarchical bootstrap: $p < 0.001$). (**C**) Joint encoding properties of glimpsed and masked target phonetic features in HG (left) and STG (right). HG mainly contains sites encoding glimpsed-only or glimpsed and masked phonetic features, while STG has more diverse joint encoding properties. (**D**) Anatomical organization of target phonetic feature encoding in STG. Masked phonetic feature encoding is negatively correlated with PA distance in STG (left). Glimpsed phonetic feature encoding is negatively correlated with distance from the pSTG-mSTG boundary (right) (ρ, Spearman correlation, $p$-values determined via hierarchical bootstrap of Spearman correlation values). (**E**) Relative encoding of glimpsed and masked target phonetic features for each electrode in the left (top) and right (bottom) hemispheres, defined as the difference in Fisher-corrected correlation between the baseline and null models. The underlying data can be found at https://zenodo.org/record/7859760. AC, auditory cortex; HG, Heschl's gyrus; mSTG, middle STG; PA, posterior–anterior; pSTG, posterior STG; STG, superior temporal gyrus.

one-sample $t$ test: $t_{(61)} = 3.79$, $p < 0.001$; bootstrap: $p < 0.001$). Together, these results show that masked phonetic features are encoded later than glimpsed phonetic features, indicating masked features require additional computation for encoding.

In addition to the difference in attentional modulation and latency of encoding between glimpsed and masked phonetic features, we also observed differences in the anatomical encoding properties of glimpsed and masked target phonetic features. To determine anatomical encoding properties, we characterized the joint encoding properties of glimpsed and masked features in HG and STG, as well as the anatomical organization of glimpsed and masked features in STG. First, we assessed joint encoding properties by determining whether individual sites in HG and STG do not encode phonetic features, encode glimpsed features alone, encode masked features alone, or jointly encode both glimpsed and masked features. This revealed whether glimpsed and masked features are encoded at the same or different sites, indicating

whether or not these features rely on the same neural substrate for encoding. On the one hand, HG mostly contains sites that only encode glimpsed phonetic features or that jointly encode glimpsed and masked phonetic features (Fig 5C, left). On the other hand, STG has more diverse joint encoding properties, mainly containing sites that jointly encode glimpsed and masked phonetic features, along with some sites that only encode either masked or glimpsed phonetic features (Fig 5C, right). Notably, the presence of sites that selectively encode masked phonetic information suggests that there exists a separate neural substrate to encode this information.

Next, we sought to determine whether the neural substrates of glimpsed and masked phonetic feature encoding in STG had the same anatomical organization. In line with the core-belt-parabelt organization of AC [38,39], prior work has shown STG responses can be predicted from HG responses [17], HG is separately connected to anterior and posterior STG [40], and neural representations grow more complex with increasing distance from primary AC [41]. Therefore, we sought to characterize phonetic feature encoding properties from the boundary between posterior STG (pSTG) and middle STG (mSTG), defined as the lateral exit point of the transverse temporal sulcus [40,42,43], coinciding with the anterolateral end of HG. We found that the degree of encoding of glimpsed phonetic features was negatively correlated (Spearman's correlation: $\rho = -0.30$, $p = 0.0198$; bootstrap: $p = 0.044$) with distance from the pSTG-mSTG boundary (Fig 5D, right), suggesting that glimpsed phonetic features may be encoded in a feedforward manner from HG. Masked phonetic feature encoding was not significantly correlated with distance from the pSTG-mSTG boundary ($\rho = -0.16$, $p = 0.21$, bootstrap: $p = 0.326$). Additionally, we characterized phonetic feature encoding properties along the posterior–anterior (PA) axis of STG. We found that the degree of encoding of masked phonetic features was negatively correlated (Spearman's correlation: $\rho = -0.42$, $p < 0.001$; bootstrap: $p = 0.025$) with PA distance in STG (Fig 5D, left). Glimpsed phonetic feature encoding was not significantly correlated with PA distance ($\rho = -0.03$, $p = 0.83$; bootstrap: $p = 0.404$). These distinct anatomical encoding properties can also be seen on a brain plot of the relative encoding of glimpsed and masked target phonetic features at each electrode (Fig 5E). Together, the joint encoding properties and organizational patterns within STG indicate that glimpsed and masked phonetic features rely on separate neural substrates for encoding.

Overall, glimpsed phonetic features are encoded in HG invariant to the listener's attention, and this representation becomes modulated by attention in STG. In contrast, masked phonetic features are encoded for only the target talker in both HG and STG. Furthermore, we find that masked phonetic features are encoded with a significant delay relative to glimpsed phonetic features at the individual electrode and population level throughout AC, indicating additional computation involved in encoding masked phonetic features. Finally, we observe distinct anatomical encoding properties between glimpsed and masked phonetic features through unique joint encoding properties in HG and STG and anatomical organizations in STG, indicating separate neural substrates for the continuous encoding of glimpsed and masked phonetic features. A video of the TRF weights of the glimpsed and masked target phonetic features over time (S1 Video) exemplifies these temporal and anatomical distinctions in encoding.

## Word onsets are selectively encoded for the target in STG

Last, we sought to determine whether lexical information is continuously represented for the target and non-target talkers. On the one hand, the observed encoding of both glimpsed and masked phonetic features for only target speech suggests that the integration of phoneme sequences into words may only occur for the target talker. On the other hand, past behavioral evidence for non-target speech processing and neural evidence for non-target phrasal structure

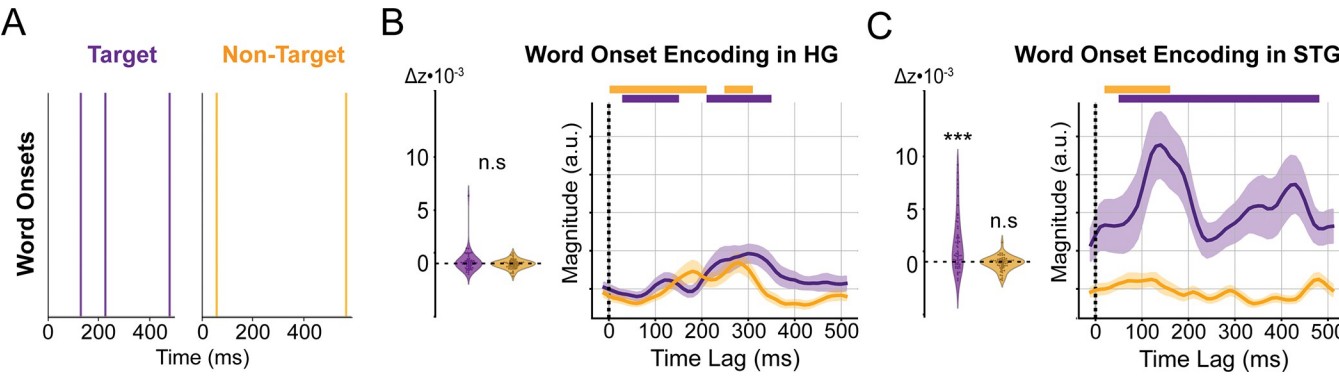

**Fig 6. Word onset features and encoding in auditory cortex.** (**A**) Example word onsets of the target and non-target talkers. The prediction correlation improvement and average TRF of electrodes in HG (**B**) and STG (**C**). TRF magnitudes are comparable across regions. Significant model improvements (***: $p < 0.001$, **: $p < 0.01$, *: $p < 0.05$) and TRF time steps (horizontal bars: $p < 0.05$) determined via hierarchical bootstrap. The underlying data can be found at https://zenodo.org/record/7859760. HG, Heschl's gyrus; STG, superior temporal gyrus; TRF, temporal response function.

encoding [44] suggests that non-target glimpses may also be integrated over longer durations. To investigate lexical encoding, we trained a word onset baseline model comprising the acoustic features (Fig 2B), phoneme onsets, phonetic features (Fig 4A), and word onsets (Fig 6A), defined as word-initial phoneme onsets. Word onsets were employed as an indicator of lexical processing because they serve as a simple and clear marker to determine whether a continuous speech stream is parsed into discrete words, as word boundaries are not clear from acoustic cues alone [45].

We measured the reduction in prediction correlation when the target and non-target word onsets were removed from the predictors. As seen in Fig 6B, we did not observe a significant encoding of word onsets in HG for the target ($t_{(65)} = 2.03$, $p = 0.073$) or the non-target ($t_{(65)} = 0.98$, $p = 0.25$) talkers. As seen in Fig 6C, we found that word onsets are selectively encoded in STG for only the target ($t_{(58)} = 5.01$, $p < 0.001$) and not the non-target ($t_{(58)} = -0.15$, $p = 0.54$) talker. Additionally, we find that the target word onset TRF has 2 peaks: an early peak occurring at 135 ms and a late peak at 425 ms. The selective word onset encoding found here agrees with non-invasive studies, suggesting that word boundaries are selectively encoded for target speech [44,46]; however, it differs in that prior TRF studies of word onset encoding only find an early peak [19]. Nevertheless, an EEG study of word onset event-related potentials showed enhanced N100s and N400s after listeners learned to segment an artificial language into words [47], suggesting that both of these components may be related to lexical segmentation.

## Discussion

Our study revealed a hierarchy of glimpsed and masked acoustic-phonetic feature encoding in AC that progressively increases in latency and attentional selectivity as we ascend the representational and anatomical hierarchy. In particular, we first found that spectrogram and glimpsed acoustic edge information is primarily encoded as the mixture, with an enhancement of target edges in STG. We also found that masked edges are encoded for both talkers in HG with increasing attentional modulation in STG. Further, we found that glimpsed phonetic features are continuously encoded for both target and non-target talkers in both HG and STG with increasing attentional modulation. Masked phonetic features are then selectively encoded for the target talker with distinct temporal and anatomical encoding properties. At the lexical level, we found that word onsets are selectively encoded only for the target talker in STG.

These results can be summarized with a model of speech encoding during multitalker speech perception, as illustrated in Fig 7. First, the acoustic mixture is rapidly encoded in HG,

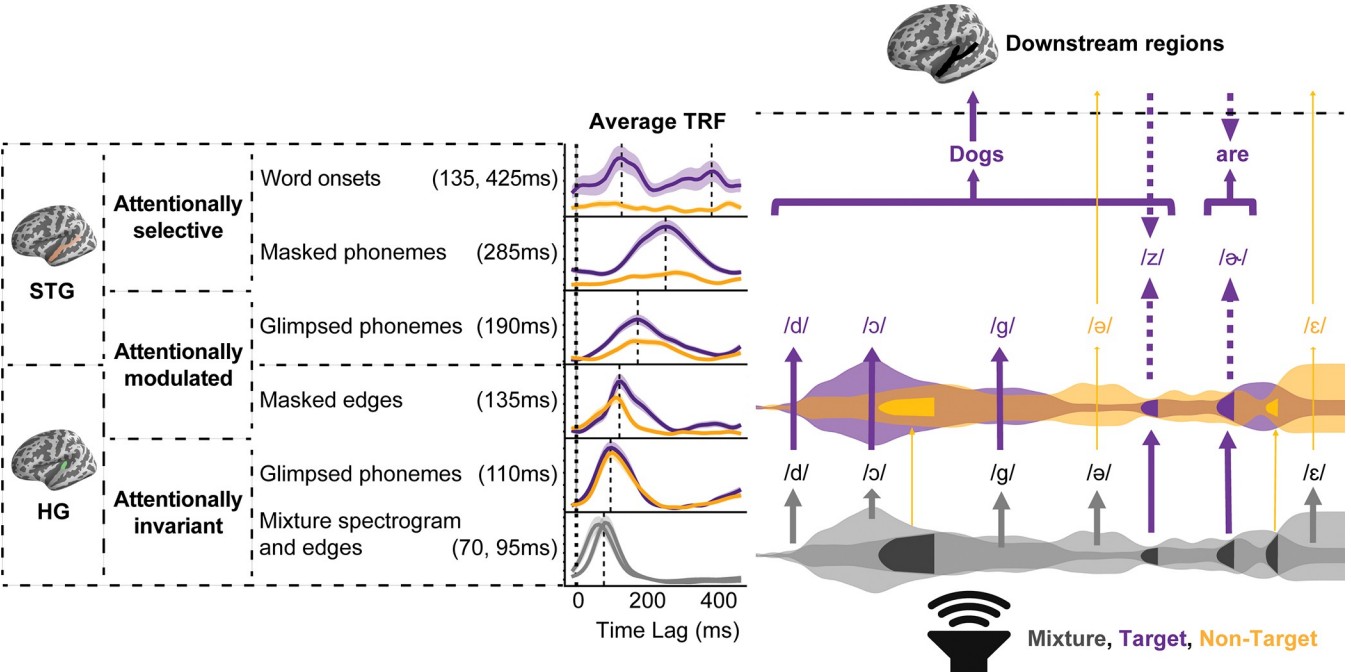

**Fig 7. Model of speech encoding in multitalker speech.** (Gray: mixture, purple: target, yellow: non-target). In the bottom row, the mixture signal, represented as 2 overlapping envelopes in gray, is first processed in HG acoustically through encoding of the spectrogram and acoustic edges. Masked acoustic edges are represented in dark gray. In the next row, phonetic information that is glimpsed in the mixture is encoded in HG invariant to attention. Both mixture acoustics and glimpsed phonetic features are encoded invariant to attention. Masked edges are then recovered in HG for both talkers with an effect of attention, showing evidence for stream segregation. The masked edges in the first row are shown to be restored with attentional modulation in the third row. In the fourth row, the encoding of glimpsed phonetic features in STG becomes modulated by attention, as indicated by the different thicknesses of arrows extending from the glimpsed phonetic features in the second row. Next, masked phonetic features are recovered in STG only for the target talker with a time relative to glimpsed phonetic features. It is unclear whether bottom-up repair (upward arrow) or top-down restoration (downward arrow) is involved in encoding masked speech. Finally, in the top row, glimpsed and masked phonetic information is integrated to form words. Beyond this point, meaning can be derived from the words of the target talker, while glimpsed non-target phonetic features may also influence the listener. HG, Heschl's gyrus; STG, superior temporal gyrus.

peaking at 70 ms for the spectrogram and 95 ms for the edges. Then, the glimpsed phonemes that appear relatively undistorted in the mixture are represented in HG around 110 ms, invariant to attention. Next, the acoustic edges that were absent from the mixture are restored in HG with an effect of attention at 135 ms—the earliest observed evidence of stream segregation. For encoding observed in STG, we see a greater influence of attention on neural representations, starting with the attentional modulation of glimpsed phonetic features at 190 ms. Finally, we observe the attentionally selective and delayed encoding of masked phonetic features at 285 ms and word onsets at 135 ms and 425 ms, demonstrating a clear division between target and non-target continuous speech processing.

## Support for the glimpsing model of speech perception

Over the past 70 years, the glimpsing model of speech perception has steadily gained support. One of the first studies of the intelligibility of interrupted speech found "that one glimpse per phoneme is sufficient" to support speech perception [23]. It was also shown that fluctuations in the envelope of a masker cause a release from masking, which was termed "listening in the valleys" [48]. Later, computational models of glimpsing showed that these models agree with psychophysical data, and both computational and behavioral performance is tightly correlated with the proportion of glimpsed speech [25].

These models view glimpsing as a low-level perceptual process that exploits momentary changes in SNR to enable auditory attention to focus on the regions of the mixture where speech is most clearly defined [49]. Notably, most methods for detecting glimpses are invariant to the glimpsed source [50,51]. Thus, glimpsing is thought to occur for both target and non-target speech. Two versions of the glimpsing model have been proposed [25]. In the glimpse-only version, glimpses alone are sufficient to support speech understanding, and this version matches behavioral data using a glimpse SNR threshold of 0 dB. In the glimpse-plus-mask version, glimpses are processed, but an additional mechanism exists to leverage regions of masked speech. This version matches behavioral data using a glimpse SNR threshold of −5 dB.

Our results support the glimpsing model as a description of the neural mechanisms of multitalker speech perception. First, we find that glimpse encoding initially occurs for both talkers in HG and later with an effect of attention in STG. The observed HG encoding is in line with past studies showing that neural responses in HG encode phonetic information [52], do so at sites that respond with a greater latency [53], and are minimally modulated by attention [17]. The observed STG encoding suggests that non-target glimpses are suppressed but not eliminated from this later stage of encoding. Additionally, the decrease in target glimpse encoding with distance from the pSTG-mSTG boundary agrees with the core-belt-parabelt organization of human AC [38,39], comprising a representation that grows more complex with increasing distance from primary AC [41]. Together, these results support the notion of glimpsing as a low-level perceptual process that generates representations upon which attention subsequently operates.

Second, we find that the optimal SNR threshold for determining what constitutes a glimpse is approximately −4 dB. While this does not exactly match the −5 dB indicated by the glimpse-plus-mask hypothesis, it similarly indicates a distinction between regions of the spectrogram that are unaffected versus distorted versus overwhelmed by the masker [37]. In this case, separate mechanisms may be needed to repair distorted regions in a bottom-up manner versus restore overwhelmed regions in a top-down manner. This hints at the possibility that masked edge encoding may relate to the repair of distorted regions, while masked phonetic feature encoding may relate to the restoration of overwhelmed regions. The difference in the attentional modulation versus attention selectivity of masked edges versus masked phonetic features supports this distinction. Nevertheless, it is also possible that the encoding of masked phonetic features represents the repair of segments not completely overwhelmed. Regardless, a glimpse threshold of −4 dB optimizes neural response prediction correlations, indicating that partially distorted acoustics can be considered glimpsed.

Third, we find that both glimpsed and masked target phonetic features are encoded with distinct temporal and anatomical patterns. In particular, we find that masked target phonetic features are encoded approximately 95 ms later than glimpsed target phonetic features both in STG and across AC. This suggests that additional computation must occur to support the encoding of masked phonetic features. We also find that glimpsed phonetic features are robustly encoded in HG and STG, while masked phonetic features are primarily encoded in STG. This is in line with the joint encoding properties of these regions, indicating HG sites encode glimpses alone or glimpsed and masked speech, while STG sites encode a more diverse set of features, including sites that encode masked speech alone. STG sites also show distinctions in the anatomical organization of glimpsed versus masked phonetic feature encoding. This suggests that glimpsed and masked phonetic features rely on separate neural substrates for computation and representation and, along with the added encoding latency, indicates that these representations are encoded with distinct mechanisms. Taken together, these results support the glimpse-plus-mask account of the glimpsing model.

## Encoding of glimpsed non-target speech

Evidence for the bottom-up account of glimpsing, including the encoding of non-target glimpses, could support a range of effects attributed to non-target speech processing. Most notably, "informational masking can be understood as arising from the incorrect assignment of glimpses to the developing speech hypothesis," such that non-target glimpses lead to interference with or competition for phonetic processing [25]. This possibility opens up new ways to model the expected interference from informational masking based on the specific non-target speech that is glimpsed. Glimpsed non-target speech may also serve to prime listeners without their knowledge. This may occur through individual phonemes or syllables, but it could also occur through words. While our evidence indicates that glimpses alone do not enable the continuous perception of non-target words, it is possible they enable the sporadic perception of non-target words. This sporadic, glimpse-based form of non-target word perception may account for priming, as well as the tracking of non-target phrase boundaries [44].

These results can also be described by the attenuation theory of attention [54], in which unattended sound sources are attenuated rather than completely eliminated. Here, glimpsed phonetic features form the specific representations that are differentially attenuated. Indeed, attenuation theory suggests that lexical processing of unattended speech can occasionally occur from glimpses alone depending on context, priming, and subjective importance. Nevertheless, our evidence is unable to adjudicate whether this automatic lexical processing can occur or if certain properties of glimpsed non-target speech draw a listener's attention to elicit momentary lexical processing and enable behavioral effects. This is why we refrain from referring to "attended" and "unattended" talkers, instead preferring the methodology-based terms of "target" and "non-target" talkers, as we cannot make any claims regarding the moment-to-moment attentional state of the listeners [55].

## Distinct temporal and anatomical encoding of masked target speech

The distinct temporal and anatomical encoding of masked target phonetic features indicates that glimpsed and masked phonetic features are encoded via distinct mechanisms; however, it remains unclear whether the mechanism used to encode masked speech operates via bottom-up repair or top-down restoration. On the one hand, it is possible that masked phonetic features are recovered by a bottom-up repair mechanism. Under this hypothesis, phonemes that are distorted in the acoustic mixture may be locally repaired. This process may take additional time due to the additional computation needed for local repair and may utilize a separate neural substrate to aid repair. However, if masked repair is a low-level mechanism, we may expect the encoding of masked phonetic features to be attentionally modulated, rather than attentionally selective, because this mechanism could not strictly distinguish between target and non-target speech in a bottom-up manner. Notably, this bottom-up repair hypothesis would also predict that phonemes that are absent from the acoustic mixture, for example, replaced with noise, would be impossible to perceive; however, the phoneme restoration effect indicates that this is not the case [26].

On the other hand, it is possible that masked phonetic features are recovered by a top-down restoration mechanism. Under this hypothesis, phonemes that are overwhelmed by the background may be recovered by a top-down filling-in. This process may take additional time due to the computation needed to predict and perceptually restored the missing phonetic information and may utilize a separate neural substrate to perform this prediction and restoration. Given that top-down restoration would involve a high-level mechanism, we would expect this process to be attentionally selective as it would operate on the ongoing speech hypothesis of a single talker. Similarly, top-down restoration would permit the perception of phonemes absent

from the acoustic mixture in accordance with the phoneme restoration effect. Notably, a previous iEEG study into the restoration of noise-replaced phonemes showed that the perceived phonemes are maximally decodable from STG around 300 ms [56], in agreement with the encoding latency of masked phonetic features seen here. This study also showed that masked phonemes are decodable from frontal regions in the period preceding phoneme onset, suggesting that bias from a predictive mechanism influences the eventual percept.

The attentional selectivity of masked phonetic feature encoding and evidence of the phoneme restoration effect suggests that masked phonetic features are restored via a top-down mechanism. In contrast, the attentional modulation of masked acoustic edge encoding suggests that masked edges are repaired via a bottom-up mechanism. Prior evidence of masked edge recovery suggests that this process occurs through a weighting of features based on their likelihood of belonging to the attended source [22], potentially through the modulation of excitability of regions tuned to the spectrotemporal properties of the target [57]. As indicated earlier, this agrees with the observed below-zero glimpse threshold, by which unaffected speech is simply glimpsed, distorted speech is repaired then glimpsed, and overwhelmed speech is separately restored. While our results could be explained through other means, this picture most clearly explains our data and suggests an interesting predictive function of attention during speech perception. Nevertheless, further studies will be needed to adjudicate between the various influences on masked feature encoding, such as masking over time versus masking over frequency [58] and the temporal coherence of glimpsed and masked cues [59–61].

Furthermore, the increased latency of masked versus glimpsed phonetic feature encoding offers interesting insight into the temporal dynamics of phonetic and lexical feature encoding. The significant difference in latency between glimpsed and masked phonetic feature encoding brings up the question of how the brain may align phonemes that are encoded via separate mechanisms of variable latency. While the acoustic cues used for phoneme discrimination are roughly stationary, neural phoneme representations evolve rapidly over time, suggesting that the brain keeps track of a short phonetic history [62]. This indicates that the brain may also keep track of missing phonetic information to seamlessly integrate its predictions into the ongoing speech hypothesis; however, a temporal generalization analysis [63] would be needed to compare the evolving representations and integration of glimpsed and masked speech.

Our observed encoding of masked phonetic features in STG at 280 ms occurs later than most other feature representations, including probabilistic phonotactics [64,65], phoneme surprisal, and cohort entropy [19]. This suggests that both phoneme statistics and the narrowing of lexical candidates may be involved in the process of restoring masked phonemes. This suggests that lexical access could occur before the encoding of masked phonetic features. While this may not be necessary for semantic understanding of speech or conceptual memory, the restoration of masked phonetic features may still serve perceptual memory [66].

## Applications

The results shown here present a novel understanding of speech perception in a multitalker environment that can provide useful insight into the difficulties faced by hearing-impaired listeners and present new possibilities for developing assistive neurotechnology to aid perception in these environments. Given that the encoding of masked phonemes appears to be distinct from glimpsed phonemes, difficulties perceiving speech in cocktail party settings may be further distinguished between difficulties with glimpse perception versus masked recovery. In particular, diagnoses for hearing-impaired listeners could be distinguished between glimpse perception and masked recovery. Therefore, difficulties with glimpse perception could be

quantified based on differences in the glimpse SNR threshold or differences in the temporal resolution used to determine the glimpse ratio [67]. Similarly, difficulties with masked recovery could be quantified, although it is unlikely age-related hearing loss is due to the inability to recover degraded speech [68,69]. Finally, the attentional modulation of neural speech tracking has previously been leveraged to build auditory attention decoding (AAD) systems that have the potential to improve hearing aids by exclusively amplifying the attended speech stream [70,71]; however, the accuracy and robustness of these systems has prevented their implementation. These results suggest novel approaches to AAD that have the potential to improve accuracy through the use of a set of features more useful for this task. In particular, it may be beneficial to decode features that are selectively encoded instead of features that are simply enhanced by attention. For example, decoding the masked phonemes of each talker may be more effective than simply decoding the spectrogram [72] because spectral information is primarily encoded as the mixture, while masked phonemes are exclusively encoded for the target talker. Nevertheless, further studies are needed to assess the efficacy of such decoding strategies.

## Limitations

This study is partially limited by the subject population, experimental paradigm, and choice of analysis. Because this study was conducted in patients who underwent surgery for drug-resistant epilepsy, their working memory capacity may differ from normal listeners. Therefore, the observed non-target encoding may result from a diminished capacity to block out distracting information [73]. Furthermore, this study was conducted with co-located male and female talkers. Previous studies have shown a release from informational masking when target and non-target talkers are different sexes or spatially separated [5,7]. This suggests that the observed non-target encoding may increase when the talkers are the same sex or decrease when the talkers are spatially separated. More experiments will be needed to see how encoding properties change in various multitalker conditions. Additionally, our behavioral task required subjects to report their perceived speech, demanding the use of perceptual memory. It is possible that masked phonetic feature encoding only occurs when perceptual memory is required. Finally, these analyses were confined to the high-gamma responses of electrodes in AC. It is known that attention also influences the encoding of speech in low-frequency bands of the neural responses, both with invasive [15] and non-invasive [70,74,75] studies. However, it is unclear how these results translate to low-frequency neural data and, thus, how they might contribute to AAD using non-invasive recording modalities. While high-gamma power in EEG has been shown to be useful for AAD, the low-frequency and high-gamma decoders were observed to have distinct spatiotemporal characteristics [76], suggesting that we may observe different encoding properties in low-frequency recordings.

While the reported latencies were computed using the peak of the average TRF of each feature, they may not be directly comparable, as acoustic and phonetic features are represented differently. On one hand, the acoustic information is represented as a spectrogram that directly relates to the physical properties of sound at each moment in time. On the other hand, the phonetic information is represented using pulses that extend from phoneme onset to offset. Therefore, it is possible that the TRF models have access to phonetic information before that information is apparent from the acoustics; however, our analysis of phoneme classification from individual time steps of the spectrogram indicates that phonemes are significantly discriminable from acoustics immediately at phoneme onset, and this discriminability peaks at 40 ms after phoneme onset. This suggests that we may have overestimated phonetic encoding latencies by, at most, 40 ms, but it is reasonable to assume that phonetic information is present

at phoneme onset. The time of phonetic discriminability may also be dependent on other properties, such as phoneme position and surprisal [77]. Nevertheless, the difference in representation only influences the relative timing between acoustic and phonetic encoding; it does not influence the relative timing between glimpsed and masked feature encoding and, thus, does not affect our claim of differential encoding between glimpsed and masked phonetic features.

## Conclusions

In conclusion, we set out to understand how attention influences the encoding of target and non-target speech by characterizing the encoding properties of glimpsed and masked acoustic and phonetic features during multitalker speech perception. First, we found that spectrogram and acoustic edge information is primarily encoded as the mixture, with an enhancement of target edges in STG. Similarly, masked edges were found to be recovered for both talkers with increasing attentional modulation from HG to STG. We then found that a glimpse SNR threshold of −4 dB optimizes neural response predictions. Next, glimpsed phonetic features are found to be encoded invariant to attention in HG and modulated by attention in STG. On the other hand, masked phonetic features are exclusively encoded for the target talker with unique temporal and anatomical encoding properties. Last, word onset encoding was only found for the target talker in STG. These findings suggest distinct mechanisms for the encoding of glimpsed and masked phonetic features and provide support for the glimpse-plus-mask version of the glimpsing model of speech perception and the attenuation theory of attention. In particular, our use of high-SNR invasive electrophysiology paired with banded normalization methods for learning TRFs allowed us to reveal these key differences in the way human AC processes multitalker speech, providing key neural evidence for existing models of speech perception and attention. Therefore, these results improve our understanding of human speech processing and present new routes to explore the understanding of non-target speech processing and the development of assistive hearing neurotechnology.

## Methods

### Human subjects

Seven subjects who were undergoing clinical treatment for epilepsy participated in this study. Five subjects were located at North Shore University Hospital (NSUH), and 2 subjects were located at Columbia University Medical Center (CUMC). Two subjects were implanted with high-density subdural electrode arrays over the left temporal lobe with coverage of STG, and one of those subjects also had a depth electrode implanted in the left AC with coverage of HG. The remaining 5 subjects had depth electrodes implanted bilaterally, with varying amounts of coverage over the left and right auditory cortices for each subject. Overall, subjects 1–6 contributed 14, 8, 3, 17, 19, and 5 electrodes to the group of 66 HG electrodes, while subjects 2–7 contributed 5, 2, 7, 3, 16, and 26 electrodes to the group of 59 STG electrodes. Further coverage information can be seen in S2 Fig.

### Ethics statement

All subjects gave their written informed consent to participate in research before electrode implantation. All research protocols were approved and monitored by the institutional review board at the Feinstein Institute for Medical Research and Columbia University Medical Center (CUMC), and all clinical investigation was conducted according to the principles expressed in the Declaration of Helsinki.

## Stimuli and experiments

Each subject participated in a multitalker experiment in which they were presented with male and female talkers matched to have the RMS intensity and with no spatial separation between them. Both the male and female talkers were native American-English speakers with an average F0 of 65 Hz and 175 Hz, respectively. The speech material consisted of stories about various topics and were recorded in-house. All stimuli were presented using a single Bose SoundLink Mini 2 speaker situated directly in front of the subject. The sound level was adjusted for each subject to be at a comfortable level.

The experiment was divided into 4 blocks. Before each block, the subject was instructed to focus their attention on one talker and ignore the other. All subjects began the experiment by attending to the male speaker and switched their attention to the alternate speaker on each subsequent block. The story was intermittently paused, and subjects were asked to repeat the last sentence of the attended talker to ensure that the subjects were engaged in the task. The stories were paused on average every 20 s (min 9 s, max 30 s). The locations of the pauses were predetermined and the same for all subjects, but the subjects were unaware of when the pauses would occur. In total, there were 11 min and 37 s of audio presented to each subject.

## Data acquisition and preprocessing

The subjects at NSUH were recorded using Tucker Davis Technologies (TDT; Alachua, FL) hardware and sampled at 2,441 Hz. One subject at CUMC was recorded using Xltek (Natus, San Carlos, CA) hardware and sampled at 500 Hz, and the other subject at CUMC was recorded using Blackrock Neurotech (Salt Lake City, UT) hardware and sampled at 3 kHz. All further processing steps were performed offline. All filters were designed using MATLAB's Filter Design Toolbox and were used in both forward and backward directions to remove phase distortion. The TDT and Blackrock data were resampled to 500 Hz. A first-order Butterworth high-pass filter with a cutoff frequency at 1 Hz was used to remove DC drift. Data were subsequently re-referenced using a local scheme whereby each electrode was referenced relative to its nearest neighbors. Line noise at 60 Hz and its harmonics (up to 240 Hz) were removed using second-order IIR notch filters with a bandwidth of 1 Hz. A period of silence lasting 1 min was recorded before the multitalker experiment, and the data were normalized by subtracting the mean and dividing by the standard deviation of this pre-stimulus period.

The data were then filtered into the high-gamma band (70 to 150 Hz), which is correlated with multiunit firing rates [78], the envelope of which is known to be modulated by speech. To obtain the envelope of this broadband, we first filtered the data into 8 frequency bands between 70 and 150 Hz, each with a bandwidth of 10 Hz, using Chebyshev Type 2 filters. Then, the envelope of each band was obtained by taking the absolute value of the Hilbert transform. We took the average of all 8 frequency bands as the final envelope and resampled this response to 100 Hz [79]. This method is commonly used in neuroscience research [80].

Electrodes were tested for speech responsiveness by calculating the effect size (Cohen's D) between the distributions of the responses during speech and silence. Electrodes with an effect size greater than 0.2—considered a small but significant effect size—were retained for further analysis, leaving 66 HG and 59 STG electrodes. In total, 188 speech-responsive electrodes in and around AC were analyzed, including those in the insula, transverse temporal sulcus, planum temporale, and middle temporal gyrus.

## Plotting electrodes on an average brain

The electrodes were first mapped onto the brain of each subject using co-registration by iELVis [81] followed by their identification on the post-implantation CT scan using BioImage Suite

[82]. To obtain the anatomical location labels of these electrodes, we used Freesurfer's automated cortical parcellation [83–85] by the Destrieux brain atlas [86]. These labels were closely inspected by the neurosurgeons using the subject's co-registered post-implant MRI. Electrodes were plotted on the average Freesurfer brain template.

### Stimulus feature extraction

**Acoustic features.** Spectrograms were computed from the raw, recorded waveforms of each talker and the sum of the 2 waveforms. The spectrograms were sampled at 100 Hz and split into 10 frequency bands logarithmically spaced between 50 Hz and 8 kHz [87]. Next, the acoustic edges of each talker and the mixture were obtained by taking the half-wave rectified temporal derivative of each spectrogram. The edge representation of each talker was split into masked and glimpsed edges using element-wise operations on the acoustic edges. Masked edges were defined as the edges that were larger in an individual talker than in the mixture: $edge_{mask} = max(edge_{talker} - edge_{mixture}, 0)$. Glimpsed edges were defined as a talker's edges that were also present in the mixture: $edge_{glimpse} = min(edge_{talker}, edge_{mixture})$. The resulting edge representations are, therefore, a separation of the original acoustic edge of each talker, such that $edge_{talker} = edge_{glimpse} + edge_{mask}$. This procedure resulted in approximately 82% of edge magnitudes assigned to the glimpsed edges and approximately 18% to masked edges. This method of separating glimpsed and masked edges is the same as prior work [22].

The shuffled acoustic representations were obtained by locally shuffling the time steps in 500 ms windows. This preserves both the local and global statistics of the representations while removing the time-locked stimulus–response relationship. This shuffling method was applied to the spectrogram, glimpsed edges, and masked edges.

**Phonetic features.** The phonetic features of each talker were obtained using Prosodylab-Aligner [87,88], which takes the speech transcript, converts each word into phonemes from the American English International Phonetic Alphabet (IPA), and performs forced alignment, returning the start and end point for each phoneme in the transcript. Each phoneme was mapped to a set of 22 phonetic features, which are a subset of those used to describe the articulatory and acoustic properties of the phonetic content of speech [89]. Namely, we used the following features: voiced, unvoiced, sonorant, syllabic, consonantal, approximant, plosive, strident, coronal, anterior, dorsal, front, back, high, low, nasal, fricative, obstruent, bilabial, labiodental, alveolar, and velar (S1 Fig). Similar phonetic feature sets have been used in a variety of studies of phonetic processing [90–93]. These features were initially binary encoded through time, such that the value of the feature is 1 for the duration of a phoneme, and 0 otherwise.

To compute glimpsed and masked phonetic representations, we first computed the glimpse ratio of each phoneme, which is analogous to the glimpse area but normalized by the duration of each phoneme [25]. This metric is computed element-wise using the 100-dimensional spectrogram and intended to quantify the proportion of time-frequency components of the spectrogram within the duration of each phoneme for which the talker's energy exceeds that of the background by a given SNR.

$$ratio_{glimpse} = \frac{|spectrogram_{talker} \geq spectrogram_{background} * (glimpse\ SNR)|}{|spectrogram_{talker}|}$$

To generate the glimpsed phoneme representations, the glimpse ratio of each phoneme was multiplied by the binary phoneme onsets and phonetic features of that phoneme.

Similarly, the mask ratio of a phoneme is defined as the proportion of time-frequency components of the phoneme for which the talker energy is less than the background by a given

SNR.

$$ratio_{mask} = \frac{|spectrogram_{talker} < spectrogram_{background} * (glimpse\ SNR)|}{|spectrogram_{talker}|}$$

As before, to generate the masked phoneme representations, the mask ratio of each phoneme was multiplied by the binary phoneme onset and phonetic features of that phoneme. Based on this definition, the glimpsed and masked phonetic feature representations sum to the original binary phonetic features.

This procedure resulted in approximately 55% of phonetic feature magnitudes assigned to the glimpsed phonetics and approximately 45% to masked phonetics when using a 0 dB SNR glimpse threshold. Using the experimentally determined −4 dB SNR glimpse threshold resulted in a 61% to 39% glimpse–mask split of phonetic feature magnitudes.

The shuffled phonetic feature representations were obtained by randomly shuffling the mapping from phoneme to phonetic feature. With this method of shuffling, each phoneme is consistently mapped to the same set of features. This ensures that the shuffled representations have articulatory information removed while retaining both phoneme identity and time information.

**Word onsets.** The word onsets features were defined as the onset of word-initial phonemes, represented with binary impulses. The shuffled word onset features were defined as the onset of a random phoneme within each word.

## Neural response prediction procedure

**Banded ridge regression.** To determine the degree of encoding of each feature along the glimpsed and masked acoustic-phonetic hierarchy, we utilized mass-univariate multivariable linear regression models using time-lagged features of the stimulus, also known as temporal response functions (TRFs) to predict the neural responses at each electrode. Stimulus features were mapped to the z-scored neural responses using time lags of 0 to 500 ms. Additional models were trained using time lags of −100 to 600 ms for plotting purposes. These TRF models were trained using the singular value decomposition (SVD) solution to banded ridge regression. Here, each input feature had its own l2-regularization parameter, lambda, which was tuned to maximize the correlation between the predicted and true neural responses [32]. We first trained models for learning optimal lambdas, then trained baseline and null models using these optimal lambdas, all of which were trained and tested using 5-fold cross-validation.

All lambda values were learned a single time and frozen for all subsequent models. In particular, the spectrogram and glimpsed edges were first jointly tuned using a polar grid search of regularization values, permitting a single SVD to be computed per ratio of lambda values while allowing the assessment of results with different scales of lambda values. This was done because studies showing larger responses to edges versus spectrograms suggest that we should jointly optimize spectrograms and glimpsed edges without bias. The lambdas for each following feature were computed one at a time, i.e., spectrogram and glimpsed edge lambdas were frozen, then masked edge models were computed sweeping over lambdas, allowing us to identify a single optimal lambda, which was then frozen. Then, glimpsed phoneme onsets models were computed sweeping over lambdas, etc.

In this formulation, a given feature always used the same lambda regardless of the model. In particular, the lambda values for each feature are as follows: spectrogram: 42.7, glimpsed edges: 13.5, masked edges: 10.0, glimpsed phoneme onsets: 31.6, masked phoneme onsets: 17.8, glimpsed phonetic features: 144, masked phonetic features: 215, word onsets: 31.6. Lambdas for glimpsed and masked phoneme onsets and phonetic features were estimated using a 0

dB SNR glimpse threshold and frozen for models utilizing different glimpse SNR thresholds. Further, these same lambda values were used to predict the normalized responses for all electrodes. This was necessary to allow the direct comparison of TRF magnitudes across electrodes, allowing interpretability of the results.

**Prediction evaluation.** We trained baseline models in which all features of interest were used to predict the neural responses. Then, one feature is removed from the baseline model, and this null model is trained to predict neural responses. A one-sided, paired $t$ test between the prediction correlations of the baseline and null models was used to determine the degree of feature encoding for each electrode. To control for the non-independence between electrodes from the same subject, we conducted a hierarchical bootstrap ($N_{boot} = 10^4$) on the prediction correlation improvements to confirm that the observed results were not due to responses found only in a subset of subjects or electrodes [34].

In particular, we trained an acoustic baseline model in which the spectrograms (mixture, target, non-target), glimpsed acoustic edges (mixture, target, non-target), and masked acoustic edges (target, non-target) were used to predict the neural responses. Then, we trained a phonetic baseline model in which the acoustic features, glimpsed phoneme onsets, masked phoneme onsets, glimpsed phonetic features, and masked phonetic features were used to predict the neural responses. Finally, we trained a word-onset baseline model in which the acoustic features, phoneme onsets, phonetic features, and word onsets were used to predict the neural responses. The encoding properties were characterized for all of these features, except for phoneme onsets, which were included to ensure that responses to phonetic features were related to articulatory information rather than timing information.

In addition, we trained shuffled models in which one feature is replaced with a shuffled version of that feature, and a new feature-shuffled model is trained to predict neural responses. For each feature, 10 shuffled models were trained, and we confirmed that any features that were found to be significantly encoded using the feature-subtracted model retained their significance using the feature-shuffled model. To determine which single electrodes significantly encoded a feature, we performed a one-sided, one-sample $t$ test between the baseline prediction correlation and the distribution of prediction correlations from the shuffled models for each electrode. This revealed whether each electrode could be predicted significantly more accurately with a true, unshuffled predictor compared to a shuffled predictor. This test was used to select electrodes for the analyses in Fig 5A and 5B and allocate electrodes for Fig 5C.

## Temporal response function analysis

To determine the time course of the encoding of each feature, the power (squared magnitude) of the TRF for each feature was taken and collapsed by averaging across features (frequencies for spectrograms, phonetic features for phonemes) and across electrodes within each region of interest. The power was used since certain sites respond positively to some features and negatively to others and certain features show positive responses in some sites and negative responses in others. Therefore, we utilized the power to summarize this information in a way that represents the degree to which different features are utilized to improve response predictions without needing to account for differences between facilitation and suppression occurring between features or sites with different responses. This method has been used in the past to summarize TRFs [17]. The latency of feature encoding in each region was determined by upsampling the average TRF by a factor of 2 and finding the peak of the upsampled TRF. Differences in latency were confirmed to be roughly consistent for different features, rather than driven by different features (frequencies, phonetic features). The shaded region around the TRF corresponds to the standard error over electrodes. The significant segments of the TRFs

(horizontal bars) were determined by computing the values of the original TRF that were significantly greater ($p < 0.05$) than the shuffled-feature TRFs at each time step using the hierarchical bootstrap ($N_{boot} = 10^4$).

## Anatomical organization evaluation

To determine whether sites in STG showed an anatomical organization of feature encoding, we measured the Spearman correlation between the position of each electrode and the prediction improvement with glimpsed and masked target phonetic features. In particular, we measured the Spearman correlation between each electrode's PA coordinate and the prediction improvement, as well as between each electrode's distance from mSTG-pSTG boundary and the prediction improvement. The significance of this correlation was tested with the hierarchical bootstrap ($N_{boot} = 10^4$) to confirm that these correlations were not driven by a subset of subjects or electrodes.

## Supporting information

**S1 Fig. Table of mapping from ARPABET phonemes to phonetic features.**
(EPS)

**S2 Fig.** Brain plots (inflated) of the electrodes from each subject plotted in the left auditory cortex (left) and right auditory cortex (right). Subjects 1–7 are in brown, blue, orange, green, pink, purple, and red, respectively. The underlying data can be found at https://zenodo.org/record/7859760.
(EPS)

**S1 Video. TRF weights of glimpsed and masked target phonetic features over time.**
(MP4)

## Author Contributions

**Conceptualization:** Vinay S Raghavan, James O'Sullivan, Nima Mesgarani.

**Data curation:** James O'Sullivan, Stephan Bickel, Ashesh D. Mehta.

**Formal analysis:** Vinay S Raghavan.

**Funding acquisition:** Nima Mesgarani.

**Investigation:** James O'Sullivan, Stephan Bickel, Ashesh D. Mehta.

**Methodology:** Vinay S Raghavan.

**Project administration:** Nima Mesgarani.

**Resources:** Stephan Bickel, Ashesh D. Mehta, Nima Mesgarani.

**Software:** Vinay S Raghavan, James O'Sullivan.

**Supervision:** Nima Mesgarani.

**Validation:** Nima Mesgarani.

**Visualization:** Vinay S Raghavan.

**Writing – original draft:** Vinay S Raghavan.

**Writing – review & editing:** Vinay S Raghavan, Nima Mesgarani.

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
