## [Editor Report · Decision Letter 0]

13 Aug 2022

Dear Dr Mesgarani, 

Thank you for submitting your manuscript entitled "Distinct neural encoding of glimpsed and masked phonetic features in multitalker speech perception" for consideration as a Research Article by PLOS Biology.

Your manuscript has now been evaluated by the PLOS Biology editorial staff, as well as by an academic editor with relevant expertise, and I am writing to let you know that we would like to send your submission out for external peer review. Please note that in addition to technical critiques, we will be looking to see enthusiasm from the reviewers' that the overall advance provided by this submission is sufficient for our journal. Your cover letter and submission nicely summarize the literature on which you are building, and we are also interested to hear what reviewers say about the potential implications of this work for broadening our understanding of attentional modulation of auditory processing as well as speech discrimination.

IMPORTANT: Before we can send your manuscript to reviewers, we will need you to complete your submission by providing the metadata that is required for full assessment. To this end, please login to Editorial Manager where you will find the paper in the 'Submissions Needing Revisions' folder on your homepage. Please click 'Revise Submission' from the Action Links and complete all additional questions in the submission questionnaire.

Once your full submission is complete, your paper will undergo a series of checks in preparation for peer review. After your manuscript has passed the checks it will be sent out for review. To provide the metadata for your submission, please Login to Editorial Manager (https://www.editorialmanager.com/pbiology) within two working days, i.e. by Aug 15 2022 11:59PM.

I apologize for the length of time it has taken for us to come to this initial decision - work travel on my part and a COVID infection on the part of our Academic Editor slowed things down. Feel free to email us at plosbiology@plos.org if you have any queries relating to your submission.

Kind regards,

Kris

Kris Dickson, Ph.D. (she/her)

Neurosciences Senior Editor/Section Manager

PLOS Biology

kdickson@plos.org

---

## [Decision Letter · Decision Letter 1]

26 Oct 2022

Dear Dr Mesgarani,

Thank you for your patience while your manuscript "Distinct neural encoding of glimpsed and masked phonetic features in multitalker speech perception" was peer-reviewed at PLOS Biology. It has now been evaluated by the PLOS Biology editors, an Academic Editor with relevant expertise, and by several independent reviewers. 

In light of the reviews, which you will find at the end of this email, we would like to invite you to revise the work to thoroughly address the reviewers' reports. Given the extent of revision needed, we cannot make a decision about publication until we have seen the revised manuscript and your response to the reviewers' comments. Your revised manuscript is likely to be sent for further evaluation by all or a subset of the reviewers.

**IMPORTANT - SUBMITTING YOUR REVISION**

*Re-submission Checklist*

*Published Peer Review*

*PLOS Data Policy*

*Blot and Gel Data Policy*

Thank you again for your submission to our journal. We hope that our editorial process has been constructive thus far, and we welcome your feedback at any time. I also apologize for the length of time that it took us to get this decision out to you. Please don't hesitate to contact us if you have any questions or comments.

Sincerely,

Kris

Kris Dickson, Ph.D., (she/her)

Neurosciences Senior Editor/Section Manager

PLOS Biology

kdickson@plos.org

REVIEWS:

Reviewer's Responses to Questions

PLOS authors have the option to publish the peer review history of their article (what does this mean?). If published, this will include your full peer review and any attached files.

Reviewer #1: Yes: Elana Zion Golumbic

Reviewer #2: No

Reviewer #3: No

Reviewer #1: In this paper, the authors recorded neural activity in HG and STG in neurosurgical patients and tested how the neural encoding of speech features (acoustic, phonetic and lexical-borders) is affected by a) the behavioral status of the stimulus as target/non-target, and b) by the severity of momentary masking. 

The sought to test neural prediction of "glimpsing" hypothesis for speech-in-noise perception (Cooke 2006 and others), according to which comprehension of masked speech (e.g. speech in noise or speech-on-speech) is afforded by the extraction of spectrotemporal features of target speech in brief periods where it is momentarily "unmasked" (i.e. the SNR changes to benefit the target speech), and these are sufficient to integrate over time and recover a fuller representation of the target speech. 

This is an ambitious and thoughtful project, that provides much needed insights into the effects of local acoustics on the neural representation of speech, across multiple levels, and with fine-grained neural resolution. Results nicely show the separate roles of HG and STG in representing different levels of information and/or applying stream segregation to a mixture of voices. They also show important differences between the neural representation of 'masked' vs 'unmasked' (glimpsed) portions of the speech, which is one of the main novelties of this work. 

General comments

Introduction:

* I found the overall topic to be timely, and the study well formulated and well executed. I think the introduction could benefit from a more theoretical discussion of the proposed interaction between low-level attributes (masking) and high-level cognitive control (attention). Many important studies are cited, but it would be good to clearly state what hypotheses are generated from the glimpsing model and whether it pertains to attention or is a more low-level "default" perceptual mechanism. 

* The authors often use the term "invariant representation of speech", but it's not clear exactly what they mean by this - is this an acoustic, phonetic, syntactic or semantic representation? Or all of the above? It would be useful to either define the term clearly or to choose a term that is less ambiguous.

Methods

* There is a large discrepancy in the number of masked vs. glimpsed edges (18% vs. 82%). So: a) how was this accounted for in the model, and b) is this a feature of many multi-talker contexts, or did the current stimuli just happen to have many periods of non-masking? Either way, it seems important to note that most of the speech was, de-facto, unmasked.. Also, perhaps it would be nice to show a demonstration of what a masked vs. glimpsed edge looks like in the spectrogram (maybe this was intended in Fig 1A, but to me this was not sufficiently clear).

* The procedure for creating the glimpsed/masked phonetic spectral-temporal regressors is explained relatively clearly, but I think it, too, would benefit from a visualization of the procedure (leading to what is shown in Fig 3A). For example, it would be useful to know what words/phonemes are actually represented in the 500-ms segment shown in Fig 3A.

* I also wonder what the relationship between 'glimpsed edges' and 'glimpsed phoneme' is, since the phonetic features seem to be more equally distributed between 'glimpsed' and 'masked' (p.20). 

* Many details are missing regarding the ridge regression procedure, including: what lambda values were ultimately chosen for each feature (in the baseline model) and were they different for each electrode or fixed per participant/all participants? Given the very large number of features, the "baseline model" has many free parameters - what optimization procedure and/or stopping rule was used to avoid over-fitting and ensuring a single solution?. Also curious whether similar/different lambda values were selected for the same feature in different baseline models (e.g. acoustic features included in the different baseline models). And/or if the TRFs for these features, trained as part of different baseline-models, retain the same latency properties. This might be an important sanity check..

Results

* The analyses are quite elaborate and although they are well-thought out, at times the wealth of terms might be confusing for the reader. The illustration in Fig 6 is a good attempt for summarizing the main results, but the text could be improved as well. One paragraph, for example, that I found "too detailed to swallow", is the last paragraph on p. 9, which talks about "sites that only encode glimpsed phonetics or that jointly encode glimpsed and masked phonetics… STG has more diverse joint encoding properties… etc.". Many of these terms are not really defined earlier, nor are any statistics done at the single-electrode level (as far as I can tell from the methods section). So, in what sense are some sites more or less "diverse" than others? This is just one example, but I feel that much of the results section can use an additional layer of clarification. 

* Latency - the different TRFs shown for encoding different speech features show a progression of latencies (even within region), that is broadly compatible with the idea that "more complex processes occur later". For example, the peak-latency for glimpsed phonemes in HG is shorter that for masked phonemes. However, I wonder how this would play out mechanistically on a continuous basis - wouldn't it be very difficult to read-out from this early region, if the latency varied so much on a phoneme-by-phoneme basis? (especially given that the time between consecutive phonemes can be shorter than some TRF latencies..). How do the authors understand the function of the TRF peak-latency in forming "invariant" representations for speech? 

* Word-onset analysis: the "shuffled" control regressor used to compare the word-onset regressor contained "the onset of random phonemes within each word". However, does this not create an acoustic confound, since by definition these will always be in the middle of an utterance whereas words will often (although not always) be preceded by a gap in the audio? Some acoustic control might be warranted here to rule out this alternative explanation.

Discussion

* The authors make one big "logical leap" in their inferences, suggesting that the earlier response to 'glimpsed phonetics' are a direct representation of the stimulus, whereas the 'masked phonetics/edges' are "restored" - implying some top-down "filling in" of the missing acoustics". However, this is pure speculation since no evidence is provided in the current data for the cause of this latency shift. A more parsimonious explanation could simply be delays in read-out due to perceptual load, rather than "restoration". I would encourage the authors to re-evaluate the strength of this argument, which clearly has critical theoretical implications.

* I am not sure I follow the authors logic on p. 14 that "informational masking may be due to incorrect assignment of non-target glimpses to the developing speech hypothesis of the target talker"? 

* The authors take the lack of a significant TRF to word-onsets in non-target speech as evidence for the lack of lexical processing / detection of word-boundaries. However, this conclusion does not really fit with the main message of the paper - i.e., that non-target speech is 'glimpsed' such that a detailed phonetic representation is formed, arguably providing the basis for previously reported effects of detection of words in non-target speech. This message should be clarified.

Reviewer #2: This paper studies neural encoding of speech in a multitalker situation. The authors refer to the glimpsing model of speech perception, whereby glimpses of individual talkers' speech in a mixture are enough bottom-up information to fill in the masked portions of speech using top-down, e.g. lexical, information. In this study, subjects are listening to two streams of speech that were mixed together. Rather than controlling the exact overlap of speech experimentally, the authors analyze the mixture post-hoc, to find glimpsed and masked portions of speech at different levels (acoustic edges, phonetic features). They then used a TRF analysis approach to see how these glimpsed and masked features are encoded in the neural responses. 

The main findings are that masked features (acoustic & phonetic) are encoded later than glimpsed features, which is in line with them having to be restored top-down. Whereas masked acoustic edges are encoded/restored for both speakers, masked phonetic features are only encoded/restored for the target speaker. I think this is an interesting finding that would deserve publication, but also a bit puzzling. There some major points that I would like to raise.

Major points:

1) Top-down or bottom-up?

If I understood the conceptual reasoning correctly, then the point of the glimpsing model is that masked speech has to be restored using top-down information. How is it that phonetic features are recovered for the target speaker only, but the acoustic edges are recovered for both speakers? In the summary figure 6, you give the example of the word 'dogs', where the acoustic evidence for the phoneme /z/ is masked. In this example the evidence for the plural marker /z/ has to come from context. If that's the case, yes, then I can recover the phoneme /z/ and the spectro-temporal edge that should have been there to cue the /z/. First of all, why would the brain even bother filling in this low-level information after the lexical access has occurred already? It seems that this would be unnecessary.

Secondly, if this were the non-target speaker and if we believe the result that the masked phonetic information (/z/) is not recovered in that case, how can the brain recover the spectro-temporal edge, which it seems to be doing? There is a discussion point in the paper that hints at a more local, low-level process to recover the acoustic edges (line 444), but I did not understand it, so some clarification would be useful. If there is a local mechanism though, to recover spectro-temporal edges, then phonetic features can in turn be constructed from that, without making use of top-down lexical information. How can we conclude from the current results that masked speech is indeed restored top-down, as you seem to be suggesting in your summary figure. Are the results more in line with some kind of local repair that enables recovery of edges for both speakers and then an attentional weighting afterwards?

2) Glimpsed/masked phonetics confounded with phonetic categories?

For a given phoneme, the glimpse ratio is defined as the percentage of time-frequency bins where the energy of the talker is higher than the background. I would guess that some phoneme categories should have higher glimpse ratio because they have higher acoustic energy on average. For example vowels are usually very high in acoustic energy, followed by other sonorants (nasals, approximants); the phonemes with lowest energy are stop consonants, they are defined by a period of silence followed by a burst. This would mean that glimpsed phonetics could be highlighting vowels and masked phonetics highlight consonants. This would be an important confound, and the neural difference (latency & anatomical differences) should be interpreted differently. Namely that they reflect a difference in phoneme encoding per se (vowels vs. consonants) rather than an effect of restoration of masked phonemes. 

In the paper, there is not enough information to exclude this possibility. Firstly, there's no comparison whether the glimpse ratio differs systematically for phonetic features (e.g. syllabic, consonantal). Secondly, the time courses in Fig. 3B/C represent only the average power of the TRFs to all phonetic features. We can imagine, however, that the strength of the masked and glimpsed TRFs differs systematically between different phonetic features, meaning that the average glimpsed TRF we see in 3 is more driven e.g. by vowel or sonorant features and the masked TRF by consonantal features. 

3) HG/STG differences robust?

The authors used a hierarchical bootstrap to control for the non-independence between the electrodes from the same subject. This is great, since it helps "confirm that the observed results were not due to responses found only in a subset of subjects or electrodes". However, they do these tests within a given electrode group, namely the HG group and the STG group. In order to interpret different results between HG and STG anatomically, in terms of a hierarchy of processing steps, we would need to make sure these differences are not due to a subset of participants contributing more to the HG or the STG group respectively, or a different electrode type (depth electrodes vs. grid). Firstly, the paper lacks a detailed description of electrode coverage for the different subjects, so it is difficult to judge how affected the results could be by different sampling. HG is probably targeted only by depth electrodes and not by the grid, does this affect the results? Secondly, there is only one place in the manuscript where anatomic differences are explicitly tested, Fig 4. And here, the authors resort to statistics that treat electrodes as independent samples (Correlations between electrode position and encoding strength). I would encourage the authors to provide more support for claims about differential processing in HG vs. STG or anatomical gradients, considering the potential problem I mentioned.

Additional points:

- Are the regularization values (lambdas) kept the same when computing the prediction improvement for a given features. Between the full model and the model where that feature is left out?

- A (supplementary) table of the phonetic features per phoneme would be very helpful.

- Line 721: sites with different response polarities. I don't understand this, since the dependent variable in the regressions is the high gamma power, which is already a rectified version of the neural signal. So what do you mean by destructive interference between sites?

- References 60 upwards are missing

Reviewer #3: This manuscript presents research aimed at examining the neural mechanisms underlying cocktail party attention. The authors record intracranial EEG from neurosurgical patients. They analyze the high gamma power of these data in different cortical regions (specifically, Heschl's gyrus and the superior temporal gyrus). In particular, the model this high gamma power activity using temporal response functions with different acoustic, phonetic, and word onset regressors. One key emphasis in the manuscript is on so-called glimpsed vs masked speech (i.e., segments of a speech stimulus that either are not or are masked by the other speech stream). They describe an interesting pattern of encoding of acoustic and phonetic speech features between masked and glimpsed speech and between target and non-target speech) - which they tie together in a nice model and very nice discussion.

Overall I found this paper simultaneously interesting and frustrating to read. My frustration mostly arises from major comment number 1 below - but also from some ambiguity and opaqueness in the methods and results sections. Meanwhile, I thought the introduction, actual results, and discussion were very interesting. Below, I list some concerns (one that I think is quite major) and queries for the authors that slightly lessened my enthusiasm over their nice discussion and model - albeit these may be answerable. 

Main comments:

1) My main concern with the manuscript was with the glimpsed/masked dichotomy. Specifically, I had three concerns. First, I just found it very difficult to get an intuitive feel for how the two regressors are calculated and what they really look like. For example, it seems to me that one could have a strong rising edge in one speech stream with flat loud activity in the same frequencies in the second speech stream. As I understand it, this would produce a strong onset in the target and a similar onset measure in the mixture, so it would be classed as glimpsed. But, in fact, it would be heavily masked .Similarly, if both streams had rising amplitude at the same time, would they then be classed as glimpsed? I don't have a good feel for this at all. I think some kind of figure and justification for the legitimacy of what you are doing here is important. I appreciate this approach has been published before, but I didn't really understand it in that paper either. Second, the whole dichotomy just feels false to me in the first place. Sure, sometimes speech is completely glimpsed. And sometimes it is heavily masked. But sometimes it is in between, no? I guess this is reflected in your regressors. But dividing the data into these two clean categories just feels overly simplistic to me. (Again, though I don't have a good feel for the regressors). So I am not sure how to interpret the results in the end. And, third, the resolution of the spectrogram is limited (as it has to be). So segments from the two speech streams could be classed as masked based on these broad frequency bands, when, in fact, they are not masked because they are are actually at different frequencies within those bands. 

2) The presentation of several of the key results was a bit ambiguous. A lot of this centered on the violin plots beside the TRFs. For example, there are some places where the non-significance of the results seems a bit implausible. Consider Fig 3B, phonetic encoding in HG for non-target speech. There seems like a clear TRF component for non-target speech here. I know the stats don't come out, but it seems likely to me that this is more an SNR issue rather than definitive evidence of absence. I also wondered about the masked acoustic encoding in HG in Fig 2B. Is this information really being separately encoded for each speech stream? Or does the masked nature of the speech in these segments literally make it impossible methodologically to determine that they are both separately encoded. Again, some of this might relate to my confusion with respect to the regressors themselves. But it may also relate to the fact that the glimpsed/masked dichotomy is not a dichotomy at all. Another example is the bottom panel of Fig 2C, where you claim that target speech is not encoded in STG (beyond the mixture). But there are significance bars for some latencies. What latencies did you use for the violin plots? Again, it seems there is signal here, but SNR might be low. I am definitely not convinced that there is enough evidence of absence to make the claim you make on lines 180-182. Same for the middle panel of Fig 2B. Indeed, this is a particularly egregious example. I'd go so far as to say that the violin plots are frankly implausible. 

3) Incidentally, some of my initial confusion over the results was just a misunderstanding on my part - I kept seeing the horizontal significance bars for the TRFs as belonging to the TRF plot just above them - when, in fact, they belong to the TRF below. I wonder could you space the figure out a bit better to make the distinction clear so that other readers don't misunderstand things the way that I did. 

Other comments:

1) I don't think you should be using the word "phonetics" the way you are using it - it generally refers to the study of speech sounds. I don't think you should talk about masked phonetics for example.

2) The last sentence of the introduction is a mouthful - up to you though. 

3) The abbreviation TRF was not defined on first usage.

4) I think you should specific "half-way rectified" in the main body of the text.

5) It is not true that reference 34 showed that responses in STG have been shown to primarily respond to amplitude change, rather than absolute amplitude. The comparison in that paper between envelope and peakEnv peakRate was blatantly unfair in terms of how the data were modeled. 

6) I don't have any great objection to the use of the word "prediction" when talking about modeling unseen neural responses…. But some people don't like it. You could consider using "we modeled high gamma" instead of we predicted high gamma. Up to you.

7) The way some of the figures are referred to is quite strange. For example, Fig 2A, bottom seems to be a reference in support of a reduction in prediction correlation. But it just refers to the spectrograms - which are mentioned much earlier in the sentence. It's unnecessarily confusing and happens a few times.

8) I don't know what "are comparable" means in the caption of Fig 2. Do you literally mean, "can be compared" - as in they are on the same axis so they can be compared. Or are you saying that they are similar?

9) I didn't understand the sentence beginning on line 252. I don't know why no significant TRF timesteps means that encoding latency differs.

10) Different latencies = different "mechanisms of encoding". Don't know what that means either - but you say it a couple of times (e.g., line 269).

11) "than" not "that" on line 282.

12) Not clear how you drew the conclusion that glimpsed phonetics are encoded in a feedforward manner on line 304 based on the negative correlation with distance from the boundary. Very opaque.

13) The word onset stuff is very compelling/cool.

14) No idea what you really mean to say by the sentence on lines 408/409.

15) An extra "and" on line 458.

---

## [Decision Letter · Decision Letter 2]

24 Mar 2023

Dear Dr Mesgarani,

Thank you for your patience while we considered your revised manuscript "Distinct neural encoding of glimpsed and masked phonetic features in multitalker speech perception" for publication as a Research Article at PLOS Biology. This revised version of your manuscript has been evaluated by the PLOS Biology editors, the Academic Editor and the original reviewers. 

Based on the reviews, we are happy to invite you to address the remaining points raised by the reviewers.

As you will see reviewer #3 raises an issue about embracing the null that should be addressed, and reviewrs #2 and #3 have problems with the definition of glimpsed and masked speech. Please address all the reviewers' issues.

Please also make sure to address the following data and other policy-related requests.

1. ETHICS STATEMENT:

Please include information about the form of consent (written/oral) given for research involving human participants. All research involving human participants must have been approved by the authors' Institutional Review Board (IRB) or an equivalent committee, and all clinical investigation must have been conducted according to the principles expressed in the Declaration of Helsinki.

2. DATA POLICY:

A) Supplementary files (e.g., excel). Please ensure that all data files are uploaded as 'Supporting Information' and are invariably referred to (in the manuscript, figure legends, and the Description field when uploading your files) using the following format verbatim: S1 Data, S2 Data, etc. Multiple panels of a single or even several figures can be included as multiple sheets in one excel file that is saved using exactly the following convention: S1_Data.xlsx (using an underscore).

B) Deposition in a publicly available repository. Please also provide the accession code or a reviewer link so that we may view your data before publication. 

Regardless of the method selected, please ensure that you provide the individual numerical values that underlie the summary data displayed in the following figure panels as they are essential for readers to assess your analysis and to reproduce it: Figures 2ABC, 3ABC, 4ABC, 5ABCDE, 6ABC, and supplementary figure SF3.

3. Please provide a blurb which (if accepted) will be included in our weekly and monthly Electronic Table of Contents, sent out to readers of PLOS Biology, and may be used to promote your article in social media. The blurb should be about 30-40 words long and is subject to editorial changes. It should, without exaggeration, entice people to read your manuscript. It should not be redundant with the title and should not contain acronyms or abbreviations.

4. We suggest a change in the title: "In multitalker situations humans encode speech perception differently depending on the relative loudness of target and background speakers"

We expect to receive your revised manuscript within two weeks. 

*Published Peer Review History*

*Press*

Sincerely,

Paula

---

Senior Editor,

pjaureguionieva@plos.org,

PLOS Biology

Reviewer remarks:

Reviewer #1: Elana Zion Golumbic

Reviewer #1: I appreciate the authors careful revision of this manuscript and their thoughtful toning-down of some of the mechanistic claims. 

The addition of the parametric analysis of SNR is also useful and emphasizes the fact that masking is never binary but is always a matter of degree - and that neural encoding of non-target speech may also vary as a function of this changing degree of masking. 

Another minor point that the authors may want to consider (for this paper or for future work) is how the temporal coherence across frequency bands plays in here - in the sense that if a particular speech feature (e.g. phoneme) is masked in only a portion of the spectrum, it make be easier to "recover" relative to features whose entire spectral-makeup is masked (or are more narrow band).

I also appreciate the added methodological details regarding model optimization. Since there are many choices and options for conducting these analyses, full transparency here is critical. 

The discussion is now more cautious and well balanced and eventhough many open questions remain, I believe that the manuscript in its current form makes a useful contribution to the growing literature attempting to 'reverse-engineer' how the brain encodes and processes concurrent speech. 

Reviewer #2: The authors addressed my specific concerns well. 

I'm happy to see that the glimpsed vs. masked difference mostly generalizes across the different phonetic categories (major point 2): this would have been a major concern for me.

It's good to see that almost all subjects contribute both to the HG and the STG groups of electrodes (major point 3) and that the analyses on anatomic gradients seem to hold taking subject dependencies into account.

Major point 1: I think the new introduction is much clearer on the theoretical scope of the current work, namely the difference between a glimpse-only and glimpse-plus-mask model. I have only a comment on the accessibility of the concepts:

The notion of glimpsed and masked edges is central to this work, but it's hard to understand when it goes into the details. After a few month since reading the first version of the manuscript, it took me quite some time to understand the different features. Especially in Fig. 2A it was very confusing to have 'glimpsed edges' for the mixture. I had to go back and forth between the methods, the main text and the figure to figure out how the features relate to each other. I realized only after a while that there is a new Supp Fig 1, with illustrations that explain this very well. In my mind, this belongs in the main text. Together with the illustrative examples in the new figure 3 (which illustrates the glimpse ratio for phonetic features), this would make the paper self-contained and accessible. That's a suggestion of course.

The new analyses on the glimpse threshold are a cool idea. As the authors mention, it suggests interesting differences between clear preserved edges, distorted edges and fully masked edges. In terms of the neural data, this is very exploratory, and should be clear from the paper. At the very least, there should be some tick-labels showing the magnitude of this effect: I assume it is small, and not statistically different (between -4dB and the original 0dB). That doesn't mean it's not meaningful, just the experiment wasn't designed to specifically test this idea. But there should be some way for the reader to compare the magnitude of this effect to the other reported model comparisons.

The supplementary video didn't work for me.

Reviewer #3: Many thanks to the authors for their efforts in responding to my previous comments. Their clarifications where helpful in many places. However, I still have some residual unease with the way in which glimpsed and masked speech has been defined. I feel that the authors and I have been speaking past each other a little on this issue, so I will have one more try at explaining my concern here.

The authors are defining glimpsed and masked speech based on edges. I understand that much. But that approach just seems deeply flawed to me. I'll revisit two examples from my previous review. The first was my previous point number 1. I will try to illustrate that point again with a silly extreme example. Imagine the masker speaker is screaming a steady perfect 1kHz tone in my ear at 110dB. Because their "speech" in that moment is flat, there is no edge. Meanwhile, the target speaker gently increases the energy in their speech from, say, 60 to 65 dB at 1kHz at that time. So there is an edge. And there will be an edge in the mixture. As such, according to the authors, this would qualify as glimpsed speech. However, given the masker is screaming in my ear, I don't see how this can possibly be construed as glimpsed speech. Can the authors please explain to me again what I am misunderstanding? The response to the previous review just says that the authors are "not clear why [I] would consider this edge to be heavily masked." How can it meaningfully be considered glimpsed if someone is screaming in my ear at 110dB? Furthermore, the method the authors are using seems to run counter to what the authors say in the abstract - specifically they say that glimpses "are spectrotemporal regions where a talker has more energy than the background". But that is not how it is being operationalized in the analysis. Can the authors please help me understand this discrepancy?

In the second example we previously discussed, the authors say that if the two streams had rising amplitudes at the same time, then both streams would be glimpsed. Again, I don't understand. If they are both rising at the same time, then they are - almost by definition - both masking each other, no? To take an extreme example again - if I was listening to two speakers saying different things, but with highly similar timing - then, according to the operationalization in the study, both of these streams would count as glimpsed? But they would clearly both be masking each other heavily. 

Again, I am sorry to say, I remain uncomfortable with the definition of glimpses and masks that is central to the whole manuscript. Neither the authors, nor the other reviewers seem unhappy with it though. So I don't want to stand in the way of publication necessarily. But I do think it is worth considering some further explication and justification of the approach here. I study these kinds of questions quite a lot. And I feel like if I am this uncomfortable with the whole thing, then others will be too. Again, maybe it is just me. If the authors think that is likely to be the case, then, again, I won't stand in the way of publication.

I still think there is a disconnect between the TRF plots and the violin plots in several of the figures. Several figures show clear deviations from zero in the TRF plots (at a narrow range of latencies). But, because the authors have chosen to assess prediction accuracy based on a broad range (0-500 ms) of latencies, the overall effect (as depicted in the violin plots) does not come out as significant. There is clearly some relevant activity in some of these cases though - according to the TRFs. Reporting no effect based on the violin plots (which are based on *too* broad a latency window) just feels like embracing the null to me.

---

## [Editor Report · Decision Letter 3]

19 Apr 2023

Dear Dr Mesgarani,

Thank you for the submission of your revised Research Article "Distinct neural encoding of glimpsed and masked speech in multitalker situations" for publication in PLOS Biology. On behalf of my colleagues and the Academic Editor, Jennifer Bizley, I am pleased to say that we can in principle accept your manuscript for publication, provided you address any remaining formatting and reporting issues. These will be detailed in an email you should receive within 2-3 business days from our colleagues in the journal operations team; no action is required from you until then. Please note that we will not be able to formally accept your manuscript and schedule it for publication until you have completed any requested changes.

PRESS

Sincerely, 

Paula

---

Senior Editor

PLOS Biology
